



# Experimental methodology and procedure for SAPPHIRE: a Semi-automatic APParatus for High-voltage Ice nucleation REsearch

Jens-Michael Löwe[1,*], Markus Schremb[2,*], Volker Hinrichsen[1], and Cameron Tropea[2]

[1]High-Voltage Laboratories, Technical University of Darmstadt, Darmstadt, 64283, Germany.
[2]Institute of Fluid Mechanics and Aerodynamics, Technical University of Darmstadt, Darmstadt, 64287, Germany.
[*]These authors contributed equally to this work.

**Correspondence:** Jens-Michael Löwe (loewe@hst.tu-darmstadt.de)

**Abstract.** Ice nucleation is of great interest for various processes such as cloud formation in the scope of atmospheric research, and icing of airplanes, ships or structures. Ice nucleation research aims to improve the knowledge about the physical mechanisms and, therefore improve the safety and reliability of the applications affected by ice nucleation. Several influencing factors like liquid supercooling or contamination with nucleants, as well as external disturbances such as an electric field or surface

defects affect ice nucleation. Especially for ice crystal formation in clouds and icing of high-voltage equipment, an external electric field may have a strong impact on ice nucleation. Although ice nucleation has been widely investigated for numerous conditions, the effect of an electric field on nucleation is not yet completely understood; results reported in literature are even contradictory.

In the present study, an advanced experimental approach for the examination of ice nucleation in water droplets exposed to an

electric field is demonstrated. It comprises a method for droplet ensemble preparation and an experimental setup, which allows observation of the droplet ensemble during its exposure to well-defined thermal and electric fields, which are both variable over a wide range. The entire approach aims at maximizing the accuracy and repeatability of the experiments in order to enable examination of even the most minor influences on ice nucleation. For that purpose, the boundary conditions the droplet sample is exposed to during the experiment are examined in particular detail using experimental and numerical methods. The method-

ological capabilities and accuracy have been demonstrated based on several test nucleation experiments without an electric field, indicating almost perfect repeatability.

## 1    Introduction

Ice nucleation and the involved physics are present in many technical fields such as atmospheric research, or the icing of aircraft, ships or structures (Cebeci and Kafyeke (2003); Makkonen (1987); Szilder et al. (2002)). Particularly in cold climatic

regions, icing due to accreting snow, ice particles or water droplets freezing on the surface of e.g. aircraft, high-voltage equipment or bridges can be a severe problem, causing interruption of operation or safety risks. Depending on the research area, several different influencing factors can affect ice nucleation simultaneously. During ice particle formation in clouds, multiple





influences like supercooling, impurities and external electric field are present and might influence nucleation (Vali (1996); Cantrell and Heymsfield (2005); Pruppacher and Klett (2010)). Besides the ice particle formation in clouds, the impact of

external electric fields is also important for technical applications like the icing of high-voltage components of power transmission and distribution systems (Farzaneh (2000, 2008); Laflamme and Périard (1998)). The shape of high-voltage insulators is specially chosen to withstand varying environmental stresses without functional failure; nevertheless, they eventually often fail due to icing. An ice layer on an insulator may serve as a conductive layer, eventually bridging the space between the weather sheds. This can result in creeping currents or even a flash-over in the worst case. Both homogeneous as well as heterogeneous

nucleation might be influenced by the electric field.

Whether or not ice particles are formed in clouds or on the surface of e.g. an insulator, is controlled by ice nucleation in the water droplets, which may be affected by the electric field. Although ice nucleation has been examined for various conditions over the last decades (Acharya and Bahadur (2018); Koop (2004); Vali (1996); Whale et al. (2015); Campbell et al. (2015); Hoose and Möhler (2012)), the effect of an electric field on nucleation is still not well understood. Existing models predict

that the electric field can either stimulate or inhibit ice nucleation, depending on the electric field strength (Kashchiev (1972)). Results reported in literature are even contradictory (Doolittle and Vali (1975); Stan et al. (2011); Pruppacher and Neiburger (1963)). However, further improvement and higher utilization of technical systems is only possible with a deeper insight into the physics involved in ice nucleation in electric fields.

During the past decades, several approaches have been developed to rationalize and automate experimental examination

of ice nucleation in order to better control the experimental conditions (Budke and Koop (2015); Stan et al. (2009); Barlow and Haymet (1995)). However, none of the developed approaches allows examination of ice nucleation in an electric field. Therefore, in the present study an experimental approach for the accurate examination of ice nucleation in the presence of electric fields is presented. The approach comprises a method for droplet sample preparation and an experimental apparatus for exposing the droplet ensemble to well-defined electrical and thermal conditions. By paying careful attention to sample

preparation and experimentation, the entire methodology aims at maximizing repeatability and reliability of the experimental results. Apart from ice nucleation studies, the approach also allows examination of any process related to freezing of droplets exposed to an electric field. Parts of the experimental method have been previously presented (Löwe et al. (2017, 2019)). However, in the present work, the entire approach, as well as its capabilities and estimated accuracy are more comprehensively demonstrated and discussed. In addition, the boundary conditions the droplets are exposed to during an experiment are carefully

examined using experimental and numerical methods.

## 2   Experimental methodology

### 2.1   Sample preparation

Since ice nucleation in sessile water droplets may be affected by a variety of factors such as the droplet size, their proximity to other droplets, and contamination on the substrate surface or in the droplet, special attention is required during sample

preparation in order to control boundary conditions as accurately and repeatably as possible. Sample preparation involves both





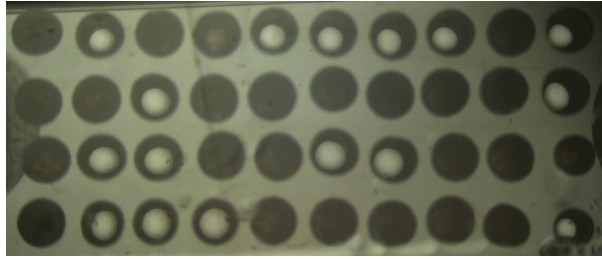

**Figure 1.** Top-view image showing a matrix composed of 40 droplets with a mean size of $d = 1.04\,$mm (wetted substrate area).

a method for generating the individual droplets and a method for depositing the droplets onto a specimen holder, with which the entire droplet ensemble may be exposed to constant conditions.

### 2.1.1 Droplet generation

Droplets of deionized and degassed water (Millipore Milli-Q Type 1, electrical conductivity $\gamma_{\mathrm{el}} = 5.5 \cdot 10^{-6}\,$S/m at $25\,°$C) are placed on a specimen holder using a piezo-driven, droplet-on-demand generator custom built according to Harris et al. (2015). The droplet size may be nominally varied through variation of the nozzle size and by adjustment of the piezo driver. However, for the present study the droplet diameter has been held constant at $d \approx 0.73\,$mm.

Although the method for droplet generation is highly reproducible in terms of droplet size, a droplet ensemble may contain outliers. Such outliers are later removed from the analysis. The maximum variation of the size of the droplets (diameter of the capped sphere on the surface) considered during analysis is typically in the range of $\pm 10\,\%$ with respect to the mean value, which is comparable to data reported in literature, e.g. $4 - 10\,\%$ of volume by Zhang et al. (2018), up to $40\,\%$ of diameter by Atkinson et al. (2016), approximately $13\,\%$ of diameter by Atkinson et al. (2013) or $3\,\%$ of volume by Budke and Koop (2015).

To minimize contamination of the droplets from the droplet generator, the entire system is purged with the same water later used for the droplet ensemble; i.e. with deionized and degassed Millipore water.

### 2.1.2 Droplet deposition

The water droplets are typically deposited onto a sapphire glass sheet with a thickness of $0.4\,$mm. Sapphire is a compromise between a substrate being electrically non-conductive, but having a high thermal conductivity ($40\,$W/(m·K) at $25\,°$C; $80\,$W/(m·K) at $-40\,°$C) to ensure a homogeneous temperature distribution throughout the substrate. The droplets are placed in an equidistant matrix using two linear stages driven and synchronized with the piezo-driven droplet generator through a custom built Arduino control program.

The contact angle of water droplets on the sapphire substrate, measured with oil as the surrounding fluid, is $\theta \approx 78°$. As a result, the droplets ($d \approx 0.73\,$mm) result in a wetted substrate area of diameter $d_{\mathrm{wet}} \approx 1\,$mm. Depending on the droplet size, a variable number of droplets can be placed on the substrate. For the current droplet size, typically 40 droplets fit on a substrate with $8 \times 15\,$mm$^2$, as shown for example in Fig. 1. Note the significantly different appearance of frozen (homogeneously dark)





80 and liquid (bright center with a dark crescent moon) droplets, which is used during analysis of the videos to automatically detect nucleation in the individual droplets. Prior to droplet deposition, the glass sheet is carefully rinsed with isopropanol. In order to minimize the generation of surface charges, any direct contact of the substrate with other instruments is avoided. Therefore, after rinsing the substrate, it is dried using only a bellows.

## 2.2 Thermal and electrical sample conditioning

85 The sapphire substrate with the sessile droplets is placed inside a ceramic body. Similar to Campbell et al. (2015), the droplets are covered with silicone oil ($\nu = 5\,\mathrm{cSt}$) to prevent their evaporation and to avoid mutual influence during the cool-down phase, e.g. via the Wegener-Bergeron-Findeisen process (Storelvmo and Tan (2015)).

The ceramic body is the central part of the experimental setup, which is schematically shown in Fig. 2. The setup comprises a system for controlling the thermal conditions for the droplets, a system for providing a well-defined electric field, and an 90 observation and measurement system.

### 2.2.1 Temperature control

The ceramic body serving as the specimen holder is made of Shapal™Hi-M Soft, which exhibits a high thermal conductivity ($92\,\mathrm{W/(m\cdot K)}$ at $25\,^\circ\mathrm{C}$) to ensure a precise and fast control of the temperature of the droplets. The same conflicting demands on the specimen holder arise as for the sample holder, and the Shapal Hi-M Soft material represents the best compromise: it 95 is electrically non-conductive while being a good heat conductor. Two Peltier elements stacked below the ceramic block and driven using a commercially available Peltier controller (Meerstetter TEC-1090-HV) are used to control the temperature of the ceramic body, i.e. the temperature of the droplets. The heat transferred through the Peltier elements is dissipated by a commercially available water cooled CPU chiller (Enermax Liqumax 240). The thermo-electric control of the ceramic temperature using Peltier elements allows a highly variable dynamic control. Any kind of temporally varying temperature profile may be 100 applied to the droplet ensemble, for example a constant cooling rate, a constant temperature or any other transient temperature profile.

Good thermal conductivity among the different parts of the system (i.e. the ceramic, the Peltier elements and the CPU cooler) is ensured through thermally conductive but electrically non-conductive thermal paste, allowing constant cooling rates of up to $5\,\mathrm{K/min}$ down to temperatures of $\vartheta_\mathrm{d} \approx -40\,^\circ\mathrm{C}$ without large deviations. Therefore, the entire temperature range relevant for 105 both heterogeneous and homogeneous ice nucleation is covered with this setup (Langham et al. (1958); Franks (1982)). Due to the characteristics of Peltier elements, the minimum temperature achievable without larger deviations from the prescribed cooling rate decreases with decreasing cooling rate.

The control electronics of the cooling system is placed below the CPU cooler. It is shielded from the electric field in the ceramic body by a grounded aluminum foil placed between the ceramic and the Peltier elements.

The experimental setup is enclosed in Styrofoam to minimize heat transfer from the surroundings. Nevertheless, during experimentation the entire setup is placed in a climatic chamber controlled at $10\,^\circ\mathrm{C}$ to maintain constant ambient conditions. The exact cooling procedure is adapted to the experiment to be performed. For example, for nucleation experiments with a



**Figure 2.** Schematic of the experimental setup used for the examination of ice nucleation in a droplet ensemble exposed to an electric field, further extended and republished with permission of SAE International, from Löwe et al. (2019); permission conveyed through Copyright Clearance Center, Inc..

constant cooling rate, the entire system is held at a constant temperature for a minimum of 5 minutes, which first allows the setup to thermally equilibrate. After that the specimen holder is cooled down with a prescribed cooling rate.

The temperature distribution on the sapphire substrate not covered with silicone oil, measured with an IR camera (Vario-Cam hr Head HiRes 640) during the cool-down phase with a cooling rate of $\dot{T} = 5\,\text{K/min}$, is shown in Fig 3. The emission coefficients of the different materials have been obtained from individual calibrations for each material. The measurement does not serve as a quantitative reference for the actual temperature measurement. However, it illustrates well the homogeneous temperature distribution over the entire sapphire substrate and all other parts of the substrate, which finally improves the

repeatability and reliability of the experimental data.

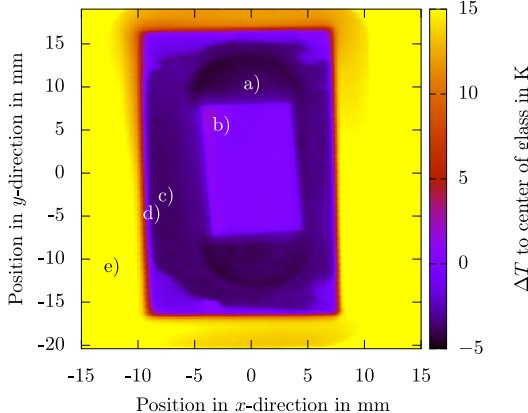

**Figure 3.** Infrared thermography image of the sapphire substrate placed inside the ceramic during the cool-down phase at a cooling rate of 5 K/min. The data represents the temperature difference at each position with respect to the temperature of the sapphire substrate in the image center (0,0). Note the homogeneous temperature distribution in all parts, which are a) bottom of the ceramic groove ($\Delta T \approx -3.8\,\text{K}$), b) sapphire substrate ($\Delta T \approx 0\,\text{K}$), c) top surface of the ceramic ($\Delta T \approx -3.7\,\text{K}$), d) sealing between PMMA cover sheet and ceramic body ($\Delta T \approx 5\,\text{K}$), and e) PMMA cover sheet ($\Delta T \approx 19\,\text{K}$).

### 2.2.2 Electric field generation

The electric field at the droplet position is generated using two electrodes embedded into the ceramic body parallel to each other and in alignment with the sapphire substrate surface. This orientation and the resulting tangentially aligned electric field is chosen to account for the associated macroscopic droplet behavior. In terms of droplet deformation and excitation, a tangentially aligned electric field has a significantly stronger effect on a sessile droplet compared to an electric field aligned normal to the surface (Sarang et al. (2011)) and it represents the typical situation on a line insulator. Accordingly, also the effect of the electric field on nucleation is expected to be higher in the case of a tangentially aligned field. To guarantee a well-defined electric potential and a partial discharge free environment in the narrow gap, the inner surface of the holes containing the brass electrodes is coated with an electrically conductive paint, and the electrodes are embedded into an electrically conductive adhesive.

The Peltier elements and control electronics below are shielded from the electric field in the ceramic by a grounded aluminum foil placed between the ceramic and the Peltier elements. The grounded foil affects the electric field distribution, which necessitates the electric field generation using two high-voltage sources of opposite polarities to generate a uniform field tangentially aligned to the glass sheet surface (Löwe et al. (2019)). A comparison of the numerically obtained field distributions (Comsol Multiphysics®) resulting from supplying only one or both of the electrodes is shown for constant boundary conditions in Fig. 4. As shown in the figure, using only one high-voltage source in combination with grounding one of the electrodes results in a significantly distorted and non-uniform electric field. Both the aluminum foil and the left electrode have the same potential and a comparable distance to the right high-voltage electrode. Therefore, the electric field lines are oriented towards





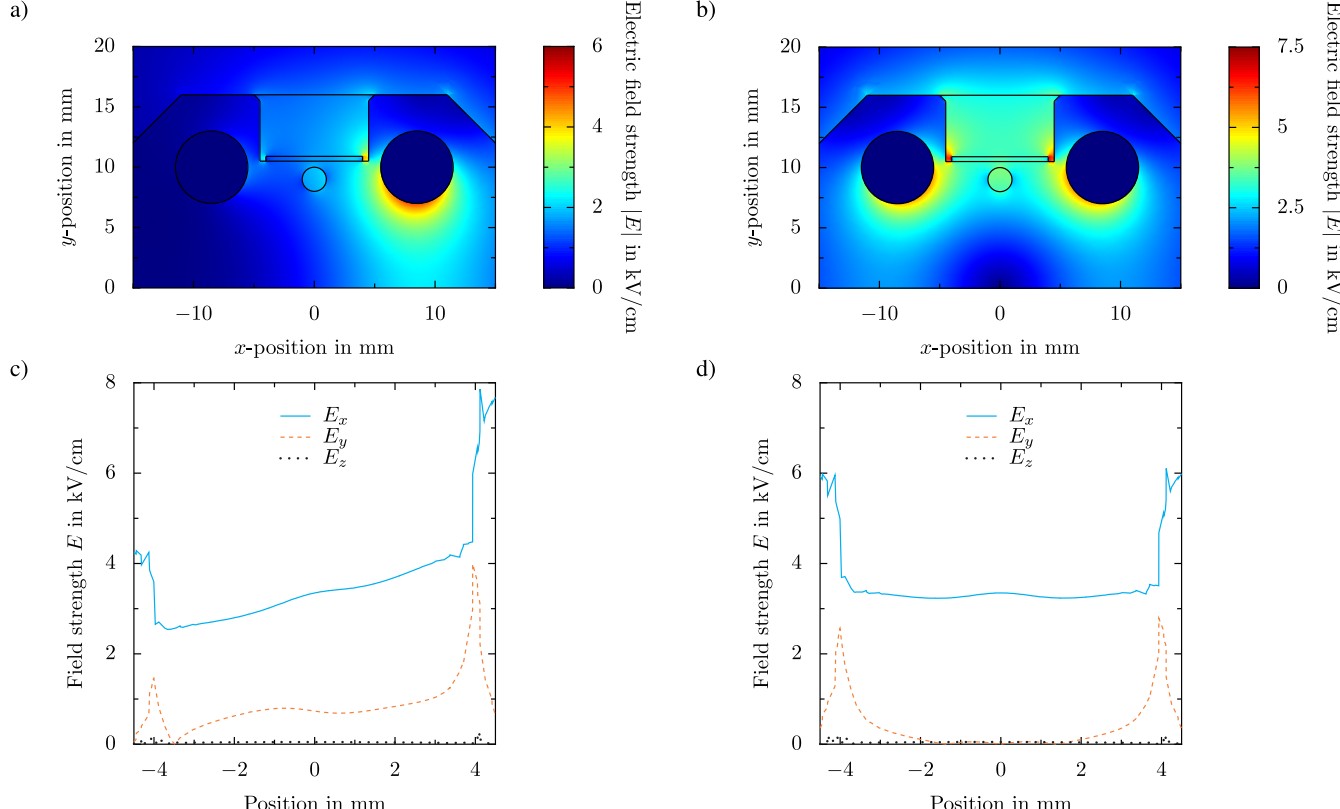

**Figure 4.** Comparison of the numerical results of the electric field distribution in the x-y-plane of the ceramic (a and b), and the different components of the electric field strength along the x-direction in the center of the ceramic at the sapphire substrate surface (c and d). In the case of a) and c) the electric field is generated using a single voltage signal of $\hat{U} = 4.0\,\text{kV}$ in combination with a grounded electrode, and for b) and d) two voltage signals of $\hat{U} = 2.0\,\text{kV}$ with opposite polarity are each used to supply one of the electrodes. In both cases, the applied voltage alternates with a frequency of $f = 50\,\text{Hz}$. Note that the field distribution is qualitatively the same for other types of electric fields.

both of the grounded surfaces and consequently, the electric field distribution is neither symmetrically nor tangentially aligned

to the substrate. To generate a well-defined electric field tangentially aligned to the glass substrate, both electrodes are supplied with separate high-voltage sources. As shown in the figure, the electric field and each of its components are then very homogeneous and almost constant inside the entire groove in the ceramic. Note that due to the characteristic time scales for charge relaxation in water, $\tau_\text{e} = \varepsilon/\gamma_\text{el}$, which is in the range of microseconds and the electric field $\tau_f$ in the range of tens of milliseconds, the water droplets on the sapphire substrate can be assumed to be perfect conductors (Löwe et al. (2020)).

Consequently, the surface of the droplets is an equipotential surface, and the electric field lines are perpendicular to the droplet surface. Nevertheless, without the droplets the electric field is tangentially aligned to the glass substrate; i.e. the electric field lines are all parallel to the substrate surface.



**Figure 5.** Schematic wiring diagram of the system for the power supply used to provide an alternating electric field. The voltage dividers are used to measure the applied voltage. The dashed rectangle represents the voltage source and can be replaced according to the required type of electric field.

Generally, three different types of electric fields can be generated with the present setup: a constant electric field, an alternating electric field with a variable frequency between $10\,\mathrm{Hz}$ and $500\,\mathrm{Hz}$, and transient electric fields, which are generated by an impulse voltage generator with a fast increase to the maximum field strength (over several microseconds) followed by a slower decrease of the field strength (over a time variable from micro- to milliseconds). The configuration of the experimental setup in terms of method for electric field generation depends on the actual type of the electric field.


**Figure 6.** Schematic wiring diagram of the system for power supply used to generate a transient electric field with high initial peak field strength and a continuous decay of the field strength.

### 2.2.3 Constant electric field

In the case of a constant electric field two identical high precision high-voltage power sources (Heinzinger PNChp 30000-2 ump) with a maximum output of $30\,\mathrm{kV}$ for each source are used. Both high-voltage sources are operated with opposite polarities and are controlled individually. The maximum applicable electric field strength is limited by the dielectric strength between the electrodes and the ground aluminum foil being prone to surface flash-overs for high voltages. The resulting maximum voltage applicable to each of the electrodes is $U = \pm 8\,\mathrm{kV}$, corresponding to an electric field strength of $|E| = 11.61\,\mathrm{kV/cm}$ in the center of the glass sheet.




### 2.2.4 Alternating electric field

The alternating electric field is generated, similar to the constant electric field, using two signals of opposite polarities provided by two high-voltage transformers, each supplying one of the electrodes. However, generation of an alternating electric field is more complex than the generation of a constant electric field. A schematic wiring diagram of the system used for the generation of an alternating electric field is schematically shown in Fig. 5. For an alternating field a function generator (GW Instek SFG-2104) and power amplifier (Thomann TA2400 MK-X) serve to generate a sinusoidal signal with a variable frequency, which is used as the input signal for the high-voltage transformers. To achieve a phase shift of $180°$ between the high-voltage signals, the same input signal is used for both high-voltage transformers with twisted phases. The applied voltages are measured individually using two custom-made voltage dividers and controlled by means of an in-house LabView program.

### 2.2.5 Transient electric field

The transient voltage is generated using two custom-made single stage Marx generators, each supplying one of the electrodes. A schematic of the wiring diagram of the power supply system is shown in Fig. 6. The characteristics of the impulse voltage can be varied through the resistors. To ensure that both Marx generators generate the same voltage, they are supplied with an alternating voltage using a single source. Using two anti-parallel diodes allows charging the Marx generators to opposite polarities. The generators are synchronized by triggering their spark gaps simultaneously, and the applied voltages are approximately adapted to standard lightning and switching impulse voltages, as commonly used for testing in high-voltage engineering IEC 60060-1.

## 2.3 Measurement system

### 2.3.1 Observation system

Ice nucleation in the droplets is observed from above using a low-speed video camera (Basler A631fc) and a high magnification lens with co-axial illumination (Navitar 12x Zoom Lens). However, optional use of a beam splitter between the lens and the camera allows simultaneous capturing of the droplet ensemble via a second camera, e.g. for high-speed imaging. Optical access to the specimen inside the Styrofoam housing is provided through several PMMA sheets incorporated into the housing, which thermally insulate the specimen from the camera lens placed above. Gaseous nitrogen in the volume between the PMMA sheets prevents condensation of water, which would obscure observation of the droplet ensemble.

The wide zoom range of the lens, in combination with the beam splitter, offers large variability of the experimental setup. For example, not only the entire droplet ensemble, but also individual droplets may be observed during their cool-down phase and exposure to the electric field. Using a high-speed video camera in addition to the low-speed camera allows evaluation of the entire history of a freezing droplet; taking place on significantly varying time scales: cool-down phase (slow), nucleation and supercooled freezing (extremely fast), and subsequent thawing (slow). An image sequence of a freezing droplet with $d = 1.02\,\mathrm{mm}$ (without an electric field) captured with a high-speed camera is shown for example in Fig. 7.





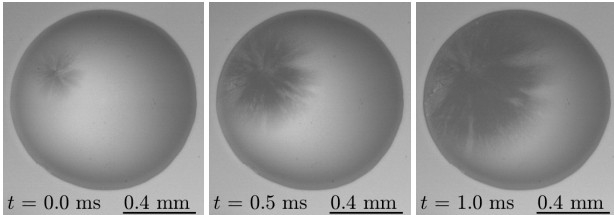

$t = 0.0$ ms   0.4 mm   $t = 0.5$ ms   0.4 mm   $t = 1.0$ ms   0.4 mm

**Figure 7.** Top-view image sequence of dendritic freezing of a sessile supercooled water droplet with $d = 1.02$ mm (wetted substrate area), captured using a high-speed video camera. Nucleation inside the droplet at a temperature of $\vartheta = -22.65\,°$C is followed by dendritic freezing of the supercooled water droplet. Note that although the shown droplet is not exposed to an electric field, the present experimental setup allows examination of that situation.

As shown in the figure, coaxial illumination and the large magnification allow detailed inspection of the processes taking place inside the droplet. In the illustrated droplet, nucleation at $\vartheta = -22.65\,°$C is followed by dendritic freezing of the significantly supercooled droplet. To the authors' knowledge, this process has never been examined before in the presence of an electric field, which is now possible with the present experimental setup.

### 195   2.3.2   Temperature measurement

The actual temperature of the droplets during experimentation cannot be measured directly due to the fact that any sensor would affect the temperature field around the droplets or even nucleation itself. Therefore, the temperature is continuously measured inside the ceramic body close to the bottom of the groove inside the ceramic. An appropriate calibration of the temperature measurement then allows indirect measurement of the temperature of the droplets during the experiments with high accuracy. 200 A fibre optical measurement system (Fiso FOT-L-SD) is used for the continuous temperature measurement inside the ceramic block, since it is a non-invasive measurement method with respect to the electric field.

During calibration the temperature difference between the droplet position (i.e. on the sapphire substrate) and the position of the actual temperature measurement during the experiments (i.e. in the ceramic block) is measured as a function of temperature for different cooling rates. For that, the temperature inside the ceramic block is controlled and the temperature on the sapphire 205 substrate is continuously measured to determine its dependence on the cooling rate and the actual temperature of the ceramic block. Temperature calibration is repeated a minimum of three times for each cooling rate, to increase the statistical significance of the calibration data. The calibration is performed under the same boundary conditions as later applied for the experiments, to maximize the accuracy of the indirect temperature measurement for the respective experiment. As mentioned before, this may involve a certain thermal conditioning of the experimental setup in the climatic chamber prior to the actual experiment. For 210 example, for experiments with a constant cooling rate, the experimental setup is kept at a constant temperature for five minutes to ensure thermal equilibrium in the experimental setup before starting the experiment, and finally, to increase the repeatability of the experimental conditions.

Typical calibration data in terms of the temperature at the sapphire substrate depending on the temperature of the ceramic is



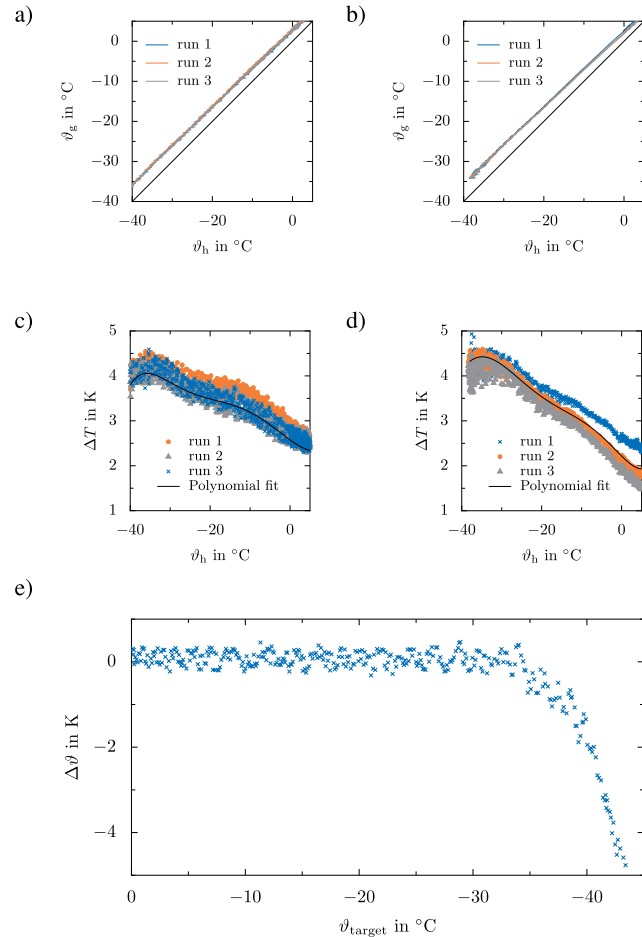

**Figure 8.** Example data obtained during temperature calibration. a) and b) Sapphire substrate temperature as a function of the ceramic temperature for $3\,\mathrm{K/min}$ and $5\,\mathrm{K/min}$. c) Difference between the sapphire substrate temperature and the ceramic temperature depending on the latter for $3\,\mathrm{K/min}$ and $5\,\mathrm{K/min}$. e) Test of the temperature calibration $\dot{T} = 5\,\mathrm{K/min}$: difference between the indirectly measured sapphire substrate temperature and the target temperature $\vartheta_{\mathrm{target}}$ prescribed by the Peltier controller in order to follow the temperature ramp, depending on the target temperature itself.

shown for a cooling rate of $3\,\mathrm{K/min}$ and $5\,\mathrm{K/min}$ in Fig. 8 a) and b), respectively. As a consequence of the thermal inertia of the system, the temperature difference between the sapphire substrate and the ceramic increases with decreasing temperature. However, since the temperature difference is mainly controlled by the thermal properties of the ceramic, the sapphire substrate and their thermal connection, the temperature difference only slightly increases with increasing cooling rate. The dependence of the temperature difference on the temperature measured in the ceramic, which is shown for example in Fig. 8 c) and d), is fitted using a polynomial function, also shown in the figures. Although the temperature difference significantly fluctuates during a calibration run and, moreover, may vary between the different calibration runs for a constant cooling rate, the trend of





the data is well described using a polynomial function of sixth degree. This relation is used as a correction for the temperature measured in the ceramic and to continuously estimate the temperature of the sapphire substrate. Finally, the temperature at the sapphire surface, indirectly measured through measuring the temperature of the ceramic, is controlled during the experiments. The resulting accuracy of the indirect temperature measurement and the associated temperature control of the test setup are illustrated in Fig. 8 e). The difference between the actual temperature at the sapphire substrate and the target temperature prescribed by the Peltier controller in order to follow a certain temperature ramp, $\Delta\vartheta$, is shown as a function of the target temperature for $\dot{T} = 5\,\text{K/min}$. Note that for this case, the temperature at the substrate indirectly measured through a temperature measurement in the ceramic is used to control the temperature. As shown in the figure, although the temperature difference obtained during calibration significantly fluctuates (compare Fig. 8 b), its estimation via a polynomial function results not only in excellent control of the sapphire substrate temperature, but also in an accurate and reliable indirect measurement of the droplet temperature during the experiment. While the maximum deviation from the desired temperature is approximately $0.46\,\text{K}$ for a target temperature down to $-34\,^{\circ}\text{C}$, the deviation increases to approximately $1.94\,\text{K}$ for a target temperature of $-40\,^{\circ}\text{C}$. Therefore, the error associated with the indirect temperature measurement increases significantly for temperatures approaching $-40\,^{\circ}\text{C}$. Thus, with a very high accuracy of temperature measurements down to approximately $-34\,^{\circ}\text{C}$, the method enables accurate examination of heterogeneous nucleation over a wide temperature range.

Note that the situation during both the calibration test run and the calibration run itself completely replicates the situation during the experiments, i.e. that the sapphire substrate placed in the groove of the ceramic is covered with silicone oil.

## 3 Discussion of the boundary conditions

Possible effects of an electric field on nucleation range from nano-scale effects involving the orientation of the dipole water molecules (Yan et al. (2014)) to macro-scale phenomena, such as droplet oscillation in the case of an alternating electric field (Reguera and Rubí (2003); Löwe et al. (2020); Schütte and Hornfeldt (1990)). Since nucleation is a physical process which is extremely sensitive to a variety of parameters, the experimental boundary conditions must be precisely controlled in order to distinguish unambiguously the effects on nucleation.

### 3.1 Oil vs. air as the surrounding fluid

In the scope of icing of high-voltage components freezing sessile droplets are surrounded by air. For the present experiments the droplets are covered with silicone oil to prevent evaporation of the droplets and to minimize mutual interaction between nearby droplets. However, both the macroscopic motion of a sessile droplet in an alternating field and the characteristics of the resulting electric field may be significantly affected by the fluid surrounding the droplet. Therefore, potential direct or indirect effects of the oil coverage on nucleation through a change of the characteristics of the electric field or the macroscopic behavior of the droplets need to be considered.

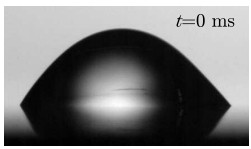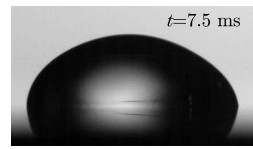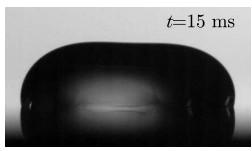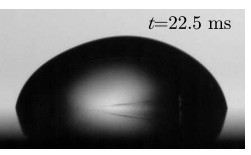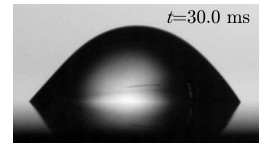

**Figure 9.** An oscillation cycle of an uncharged drop (second resonance mode) with a volume of $V = 50\,\mu l$ surrounded by air and exposed to an electric field with a frequency of $f = 17.56\,\mathrm{Hz}$ at a field strength of $\hat{E} = 3.29\,\mathrm{kV/cm}$. Similar to the present experiments, the electric field is oriented tangentially to the substrate. The oscillation is qualitatively the same as in the case of a droplet surrounded by oil. Only the amplitude of oscillation is significantly reduced, which is why it is not shown here. Videos of oscillating droplets surrounded by air and oil can be found as Supplementary Material at Löwe (2020b, a) and used under CC BY 3.0 DE.

### 3.1.1 Droplet motion

The motion of a sessile water droplet may be significantly affected by an electric field. An alternating field results in droplet oscillation depending on the droplet volume, the droplet charge, as well as the frequency of the applied electric field (Löwe and Hinrichsen (2019); Löwe et al. (2020)), and finally also the properties of the surrounding fluid. Not only the amplitude and frequency, but also the oscillation mode of a sessile droplet may significantly vary for varying boundary conditions, e.g. electric field strength, charge or droplet size (Löwe et al. (2020)). Depending on these parameters, various oscillation modes ranging from droplet oscillation mainly parallel to the surface to oscillations perpendicular to the surface or mixed oscillation modes are possible (Löwe et al. (2020); Schütte and Hornfeldt (1990)). Each of these modes is associated with a different interaction between the moving fluid molecules and the substrate where nucleation is predominantly expected.

In contrast to air as the surrounding fluid, droplet motion in oil is significantly damped due to the higher viscosity of oil ($\eta_{\mathrm{oil}}/\eta_{\mathrm{air}} \approx 252$). Figure 9 shows the motion of a droplet surrounded by air and oscillating in the second resonance mode (oscillation predominately normal to the surface). The principle direction of the motion is vertical to the substrate and can be characterized by means of the variation of the droplet height while the contact line remains pinned. An image sequence of droplet oscillation in oil is not shown in Fig. 9, since its amplitude is significantly smaller than that of the oscillation in air; thus it is not easily detected in the images. Instead, the temporal evolution of the normalized droplet height for oil and air as the surrounding fluid is shown in Figure 10 for comparison. A comparison of droplet motion shows that the principle motion and its frequency are independent of the surrounding fluid. The normalized height is defined as $z^* = z(t)/z_0$, where $z(t)$ is the time dependent height of the droplet and $z_0$ is the height of the droplet being in equilibrium without the electric field. As shown in the figure, the frequency of droplet motion is almost perfectly the same, but the amplitude of the oscillation is significantly smaller in the case of a droplet surrounded by oil ($\hat{z}^*_{\mathrm{air}}/\hat{z}^*_{\mathrm{oil}} = 5.35$). Droplet motion is drastically damped due to the higher viscosity of the oil compared to air, but qualitatively the same.

The resonance frequency of a liquid sphere embedded into another liquid is given by (Lamb, 1895)

$$f^2 = n(n+1)(n-1)(n+2)\frac{\sigma}{[(n+1)\rho_{\mathrm{water}} + n\rho_{\mathrm{surr}}]r^3}, \tag{1}$$

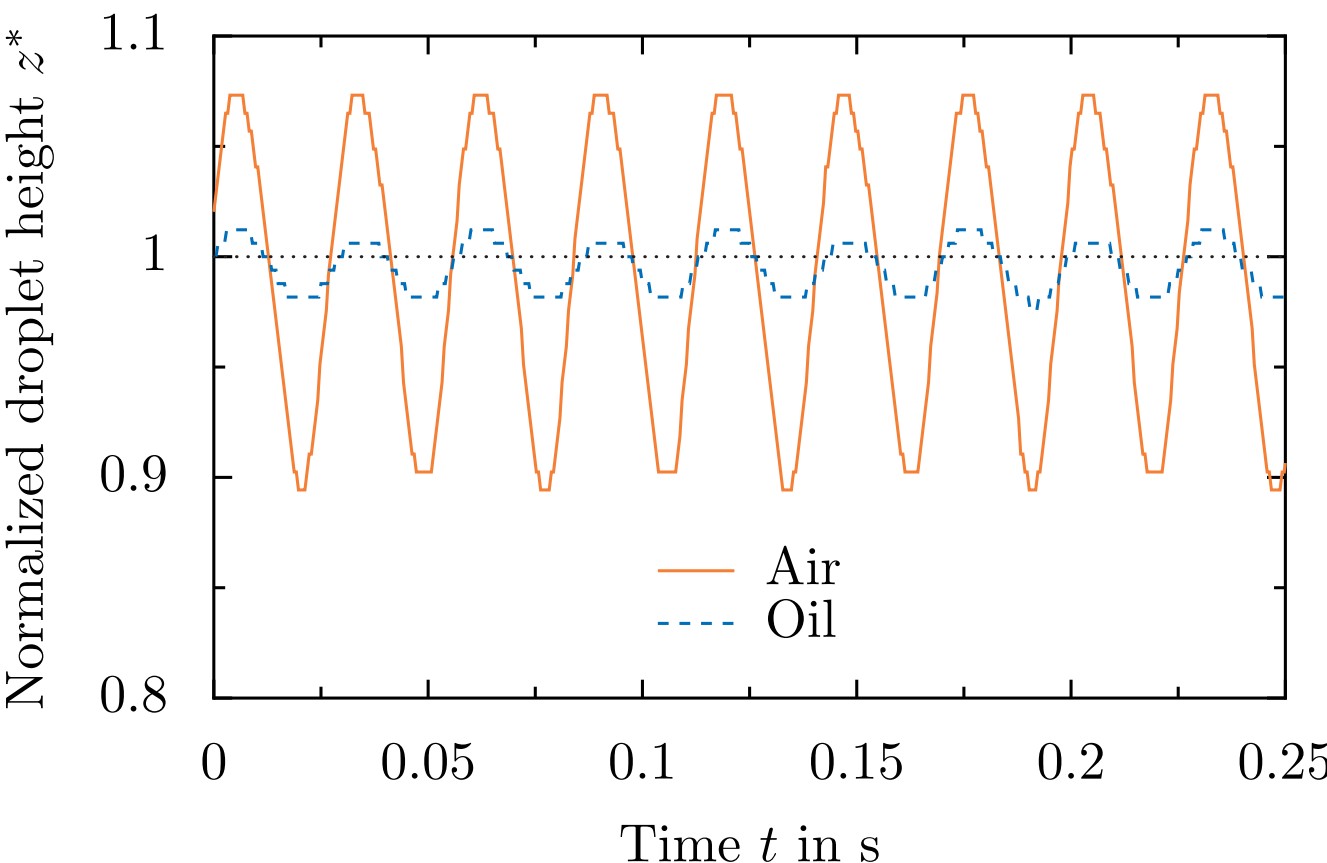

**Figure 10.** Comparison of droplet oscillation for different surrounding fluids by means of the temporal variation of the normalized height of the droplet. The data corresponds to the droplet shown in Fig. 9.

where $f$ is the resonance frequency, $n$ is an integer corresponding to the mode number, which is defined as $n-1$, and $\rho_{\text{water}}$

and $\rho_{\text{surr}}$ are the density of water and the surrounding liquid, respectively. $\sigma$ denotes the surface tension between water and the surrounding fluid, and $r$ is the radius of the droplet. Note that the resonance frequency of a sessile droplet resting on a hydrophobic surface is the same as the resonance frequency of a free droplet (Schütte and Hornfeldt (1990)). Taking into account the change of the surface tension and the density accompanying a change of the surrounding fluid from air to oil, the different resonance frequencies relate as follows

$$\frac{f_{\text{air}}}{f_{\text{oil}}} = \sqrt{\frac{\sigma_{\text{water}-\text{air}}}{\sigma_{\text{water}-\text{oil}}} \left[ \frac{(n+1)\rho_{\text{water}} + n\rho_{\text{oil}}}{(n+1)\rho_{\text{water}} + n\rho_{\text{air}}} \right]}, \tag{2}$$

$$f_{\text{oil}} = 0.55\, f_{\text{air}} \quad \text{for } n = 2,$$

for which the material properties have been assumed as $\rho_{\text{air}} = 1.204\,\text{kg/m}^3$ and $\rho_{\text{oil}} = 920\,\text{kg/m}^3$ at a temperature of $\vartheta = 25\,°\text{C}$, and $\sigma_{\text{water}-\text{air}} = 72.75 \cdot 10^{-3}\,\text{N/m}$ and $\sigma_{\text{water}-\text{oil}} \approx 35.9 \cdot 10^{-3}\,\text{N/m}$ (Peters and Arabali (2013)). Since this relation is only





valid for superhydrophobic surfaces, its validity is limited for the presently used sapphire substrate, for which the contact angle
has been measured for both air and oil as the surrounding fluid, $\theta \approx 78° \pm 7°$. As seen from Eq. (2) the resonance frequency is
decreased for all resonance modes in the case of oil as the surrounding fluid. Even if this relation is not quantitatively applicable
to the present case, the qualitative effect of oil as the surrounding fluid on the resonance frequency is presumably the same: the
resonance frequencies are expected to be smaller for the different resonance modes. The oscillation modes themselves are not

affected by the surrounding fluid, as verified by observing droplet oscillation in oil inside the ceramic for different frequencies.
Similar to the droplets surrounded by air, the oscillation is characterized by a motion parallel or perpendicular to the glass
substrate or a combined oscillation in both directions.

     Although the different resonance frequencies are shifted in the case of oil as the surrounding fluid, the actual oscillation
frequency of the droplet only depends on the excitation frequency of the electric field, as shown in Fig. 10. The surrounding

fluid only has an impact on the amplitude of the oscillation, but not on the actual oscillation frequency of the droplet. Shear in
the sessile droplets through a forced oscillation may affect ice nucleation (Reguera and Rubí (2003); Borzsák and Cummings
(1997)). Therefore, a change of the surrounding fluid, which is accompanied by a change of the situation in terms of amplitude,
could indirectly affect ice nucleation. However, increasing the electric field strength still results in an increased amplitude of
droplet motion for the same surrounding fluid and electric field frequency. Therefore, similar to droplet motion itself, also

potential effects of droplet motion on ice nucleation are only expected to be damped, rather than being fundamentally changed.

     Finally, experimental results obtained with oil as the surrounding fluid may underestimate the effect of the electric field via
droplet oscillation in comparison to its effect on sessile droplets surrounded by air.

### 3.1.2   Electric field distribution

The electric field distribution in a system of substrate, sessile droplets and surrounding fluid significantly depends on the electric

properties (permittivity and conductivity for alternating fields and constant fields, respectively) of all materials. In the case of
an alternating electric field, the ratio of the permittivities determines the field distribution. The electrically most critical region
in the system of a sessile water droplet is at the three-phase contact line of the droplet, because the electric field is suppressed
inside the droplet (due to its high permittivity of $\varepsilon \approx 80$) and is generally enhanced in the media with the lowest permittivity,
which is in most applications the surrounding air. The numerically obtained (Comsol Multiphysics®) field distribution around

two droplets deposited onto a sapphire substrate and surrounded by air and oil respectively, is shown for example in Fig. 11.
The field distribution has been obtained for identical boundary conditions in terms of electric field generation applied for
both droplets. The droplets are placed between two plate electrodes with a distance of 2.5 mm. A voltage of $U = 10$ kV is
applied between the plates, resulting in an electric field with $\hat{E} = 4$ kV/cm, which is tangentially aligned to the substrate in
the undisturbed case, i.e. without a droplet. The entire geometry replicates the real situation during the present experiments.

The droplet geometry is assumed as a rigid hemisphere; drop oscillation is not accounted for in the simulations. As shown
in the figure, the resulting field distribution is almost not affected by the surrounding fluid. However, due to $\varepsilon_{\text{oil}}/\varepsilon_{\text{air}} > 1$, the
maximum field strength is generally higher in air compared to that in oil. The small difference in permittivity between air and
oil is insignificant compared to the permittivity of water, resulting in highly comparable electric field distributions. In both





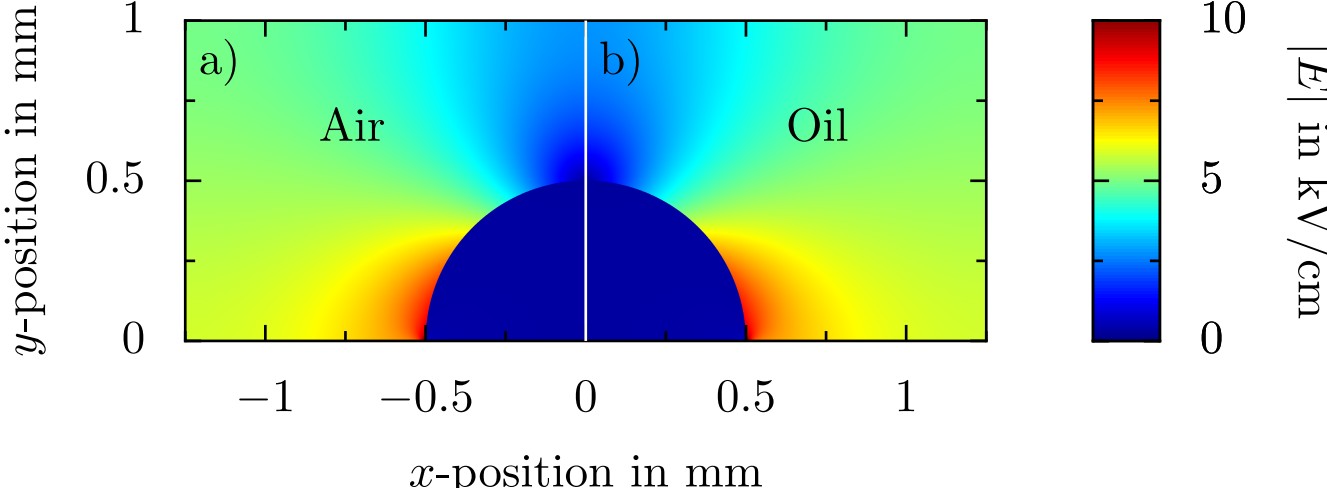

**Figure 11.** Comparison of the electric field distribution for a droplet deposited onto a rigid substrate between two plate electrodes for a) air and b) oil as the surrounding fluid. An alternating voltage of $U = 10\,\text{kV}$ ($f = 50\,\text{Hz}$) is applied between the electrodes, resulting in $\hat{E} = 4\,\text{kV/cm}$.

cases the electric field almost completely vanishes inside the droplet and is drastically enhanced at the contact line. For clarity
purposes, the field strength is limited to $|E| = 10\,\text{kV/cm}$ in Fig. 11.

As shown in Fig. 11, the most critical points of a sessile droplet prone to partial discharges are located at the three-phase contact line. Due to a singularity at the contact line, the field strength theoretically is infinite. In the case of a droplet surrounded by oil, the electrical breakdown strength of the surrounding fluid is higher than for air. As shown in Fig. 11, the electric field strength inside the droplet is not affected by the surrounding fluid. Hence, similar to its effects on droplet motion, the surrounding oil
does not alter the fundamental electrical phenomena, but significantly damps the influence of the electric field on the system.

In conclusion, although the surrounding oil may change the actual electric field in and around the droplets in comparison to droplets surrounded by air, neither dynamic motion nor the electric field characteristics are principally altered with oil as the surrounding fluid. However, due to the damped amplitudes of motion, the effect of the electric field on ice nucleation via droplet oscillation or the electric field characteristics of the system may be underestimated in the present experiments compared
to its effect on droplets surrounded by air.

## 3.2   Charges on droplets and the substrate

In the scope of electro-freezing, also electric charge may have a direct or indirect effect on nucleation. Charge on the droplets or the substrate may originate from droplet generation and sample preparation. It can result in an inhomogeneous surface charge distribution, which for example evokes an altered equilibrium shape of a sessile droplet. In addition to the static behavior of
a droplet, charge may also affect the dynamic droplet behavior. Besides the droplet volume and the electric field strength, excitation of sessile water droplets through an alternating electric field is significantly affected by the net charge on the droplet



(Löwe et al. (2020)). A highly charged droplet can change its oscillation frequency depending on the electric field strength and therefore, fluid motion inside the droplet, which can affect nucleation, is changed.

Charge is the only boundary condition, which is not actively controlled during the experiments, thus possibly affecting nucleation in an unknown way. However, the method employed for sample preparation aims at not affecting the charge on the substrate; in particular, direct contact with the substrate is avoided during sample preparation and handling. However, even if the charge on the droplets may vary, e.g. for samples generated on different days or by using a different source of water, a variation of the charge on the different droplets of the same sample is presumably negligible. Thus, although charge is not actively controlled in the scope of the present methodology, an unknown relevant effect of charge on the results obtained from experiments performed using the same droplet ensemble can be completely ruled out.

However, even uncharged droplets might be affected by a strong electric field. The electric field may result in large droplet deformation, eventually promoting the formation of a Taylor cone, which is accompanied by the ejection of small droplets (Taylor (1969); Macky (1931)). A continuous decrease of the droplet volume through the ejection of small droplets may cause a decrease of the wetted surface area, which generally results in a reduced drop freezing rate. Large droplet deformation is observable in the videos, allowing the applied electric field strength to be limited in order to prevent the formation of a Taylor cone. In any case, if a Taylor cone is observed during the experiments, the respective droplet is not considered in the further analysis.

In summary, an effect of uncontrolled droplet or surface charges on nucleation is negligible with the present methodology. Except for charges on the droplets or the substrate, all other experimental conditions are well controlled, resulting in a system for investigation of ice nucleation with a high level of accuracy and repeatability.

## 4 Discussion of the methodology

The capabilities and advantages of the present methodology are demonstrated and discussed on the basis of typical results of nucleation experiments. The data has been obtained for a constant cool-down phase without an electric field and is shown in Fig. 12. The data here only serves for demonstration purposes and has already been published in Löwe et al. (2019). Serving as a reference, experimental data obtained from four repetitions with $\dot{T} = 5\,\mathrm{K/min}$ and droplet ensembles freshly generated for each of the runs is shown in Fig. 12a. Experimental data obtained by employing the present methodology, i.e. by using the same droplet ensemble for the four repetitions, are shown in Fig. 12b. The experimental data is shown in the form of survival curves, descriptively illustrating the decay of the number of liquid droplets for a decreasing sample temperature. In a droplet survival curve the relative number of liquid droplets of a droplet ensemble, $N_{\mathrm{liq}}/N_0$, is shown as a function of temperature. Here, $N_{\mathrm{liq}}$ and $N_0$ are the number of droplets in the sample remaining liquid at a certain temperature and the total number of droplets in the sample, respectively.

The developed methodology aims at experimental conditions well-controlled in terms of droplet ensemble preparation and experimental conditions imposed on the sample. Despite the careful preparation the resulting survival curves may significantly





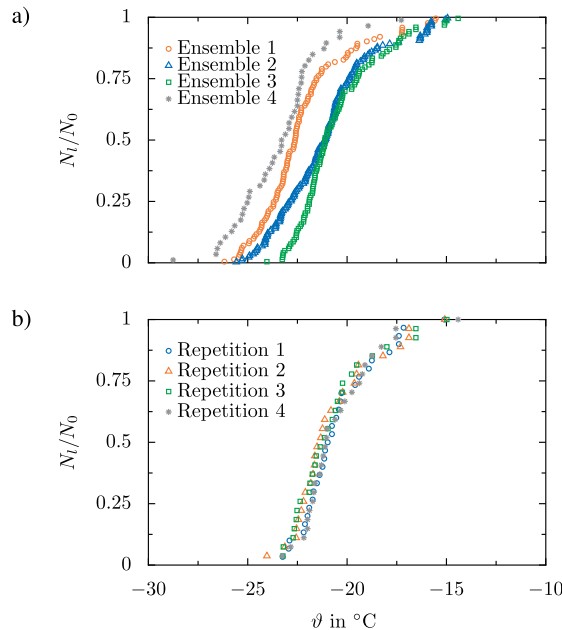

**Figure 12.** Demonstration of the present methodology by means of a comparison of the droplet survival curves obtained from several test runs without an electric field, performed using a) four different droplet ensembles, and b) the same droplet ensemble for each of the four test runs. All data is obtained at $\dot{T} = 5\,\text{K/min}$. Republished with permission of SAE International, from Löwe et al. (2019); permission conveyed through Copyright Clearance Center, Inc..

370  vary between test runs performed at constant conditions. This is especially true when using droplet ensembles freshly generated for each run, as shown in Fig. 12 a. While the characteristic S-shape of each survival curve is preserved, the temperature range, in which all droplets of a sample freeze, significantly varies for the different repetitions. Moreover, the median freezing temperature, $\vartheta_{0.5}$, corresponding to $N_{\text{liq}}/N_0 = 0.5$, significantly varies among the different droplet ensembles, $\Delta\vartheta_{0.5} \approx 2.2\,\text{K}$.

Assuming a singular nucleation model, heterogeneous nucleation in a droplet is presumed to take place at a constant temper-
375  ature associated with the ice nucleating ability of the most effective nucleation site in contact with the supercooled liquid (Vali (2014)). Nevertheless, nucleation is a stochastic process resulting in a natural variance of the results, even for perfectly constant conditions (Vali and Stansbury (1966)). Although care is taken during sample preparation, contamination of the water used to generate the droplets cannot be completely ruled out. Moreover, the ice nucleating ability of the substrate surface may also vary over the substrate. Accordingly, between two deposited droplet ensembles, both contamination of the individual droplets and
380  the substrate's influence on nucleation in the individual droplets may vary. Consequently, the effect of a parameter intentionally varied in order to examine its influence on nucleation may be masked by the variation of nucleation associated with using a fresh sample. Since little is known about the effect of the electric field on ice nucleation, which may be only small, such drastic scatter of the experimental results would necessitate an inordinately high number of repeated experiments in order to derive statistically significant results concerning the electric field effect on nucleation. Indeed, the large scatter of the data shown in





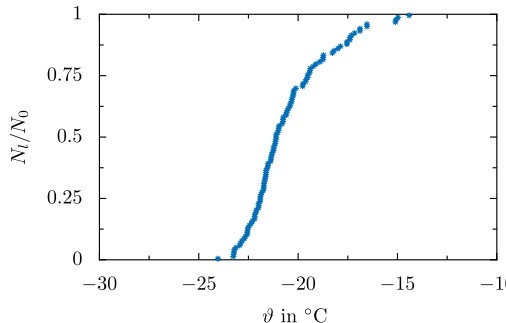

**Figure 13.** Droplet survival curve for a constant cooling rate of $\dot{T} = 5\,\text{K/min}$ and without electric field strength, derived through merging the survival curves obtained from several test runs at the respective field strength, resulting in a minimum of 83 droplets per shown survival curve.

Fig. 12 a can be almost completely attributed to variations during sample preparation, which is supported by the experimental data shown in Fig. 12 b.

As shown in the figure, when the same droplet ensemble is used for the different repetitions at constant conditions, the resulting survival curves collapse almost perfectly, confirming the singular nature of nucleation. The median freezing temperature varies by only $\Delta\vartheta_{0.5} \approx 0.35\,\text{K}$, i.e. one order of magnitude less than in the case of fresh droplet ensembles, indicating excellent repeatability of the experiment. The remaining small scatter of the survival curves can be attributed to the stochastic nature of nucleation.

Although a droplet ensemble is covered with silicone oil, droplet evaporation is not entirely suppressed. Water vapor continuously diffuses through the surrounding silicone oil, resulting in a slow but continuous shrinkage of the droplets. Nevertheless, depending on the specific experiments performed, a droplet ensemble comprising 40 droplets with a diameter of about $d_{\text{wet}} = 1\,\text{mm}$ of the wetted surface area can be used over approximately two to three days while still resulting in enough droplets with a size in the range of $\pm10\,\%$ around the mean size. Assuming four repetitions for constant conditions and approximately 40 single experimental runs feasible per day, a droplet ensemble may be used for the investigation of up to 25 - 30 different combined full experimental sets, throughout which an influential parameter may be extensively varied to elucidate its effect on nucleation.

The different methods for electric field generation have been comprehensively tested, and all work properly. Based on preliminary tests for a constant, an alternating and a transient field, the maximum possible voltage applied to each of the electrodes is $\pm8\,\text{kV}$. According to the numerical simulations (see Fig. 4), the electric field distribution is very homogeneous in the relevant region of the experimental setup, i.e. in the groove of the ceramic. All components of the electric field strength are almost constant, resulting in a well-defined electric field tangentially aligned with the substrate surface.



## 5 Conclusions

An experimental approach for the examination of nucleation in supercooled water droplets exposed to strong electric fields has been developed. Droplets of well-defined size are accurately deposited onto a supporting substrate employing an automated piezo-driven droplet generator. The substrate comprising the droplet ensemble is placed into the main experimental setup, where the sample is exposed to well-controlled thermal and electrical conditions, while being visulaized in a top-view using a video camera. Both the thermal and electrical boundary conditions are variable over a wide range. Any kind of temperature profile – constant temperature, constant cooling rate, or any other transient temperature evolution – with rates of change of the temperature of up to $5\,\mathrm{K/min}$ down to a temperature of $-40\,^{\circ}\mathrm{C}$, may be imposed onto the droplet ensemble. The sample may be exposed to a well-defined electric field tangentially aligned with the substrate surface. Not only a constant electric field, but also alternating or transient fields with a high peak field strength and a following slower decay of the field strength are possible. The maximum voltages applicable to each of the electrodes for the different types of an electric field are $\pm 8\,\mathrm{kV}$.

By using a zoom lens with a wide range of magnification, either a complete droplet ensemble or an individual droplet may be observed during exposure to well-defined conditions. Therefore, the present methodology not only offers examination of the statistics of ice nucleation in a complete droplet ensemble, but it also allows investigation of the freezing process in an individual droplet for variable conditions. Illumination of the scenery via coaxial illumination through the lens enables observation with a high contrast and excellent optical access to the processes taking place inside an individual droplet. While a slow video camera is commonly used to capture nucleation in the droplets, a high-speed camera is used to examine droplet freezing. By using a beam splitter both cameras may be operated simultaneously, e.g. in order to temporally resolve a process with both slow and fast sub-processes.

The boundary conditions typically present during the experiments have been analyzed in detail to determine their influence on the measurement result. During the experiments, the droplets are surrounded by silicone oil to prevent droplet evaporation and the interaction of nearby droplets. Even if in practice the droplets are often surrounded by air, the actual behavior of the droplets in terms of droplet motion (oscillation mode) as well as the electric field distribution in and around the droplets is still similar. However, the higher viscosity of the surrounding oil causes lower oscillation amplitudes of the droplets but no change in frequency.

The electric field around the droplets mainly depends on the properties of the surrounding fluid (e.g. the permittivity in case of alternating fields). Due to the fact that water has a high permittivity and is highly conductive, the electric field is suppressed inside the droplets compared to the surrounding. In contrast, the field is enhanced directly at the contact line. In air, the field enhancement is larger compared to oil, but the general field distribution is preserved. Hence, using oil as the surrounding fluid may cause underestimation of the effect of the electric field on ice nucleation compared to the case of air as the surrounding fluid. In addition, it is shown that the charge of an individual droplet and the substrate is negligible due to the fact that the same droplets are used for one experimental set. Except for the charges on the substrate and on the droplet, all other experimental conditions are well controlled during the experiments.





The capabilities of the entire procedure in terms of the repeatability and accuracy of the experimental results for ice nucleation has been demonstrated based on survival curves obtained from repeated experiments without an electric field. Although the entire method for sample preparation aims at maximizing the repeatability of the experiments, the test experiments indicate a significant stochastic effect of sample preparation on ice nucleation. For experiments performed at constant conditions but using droplet ensembles freshly generated for each experimental run, the resulting survival curves scatter significantly, which would then necessitate a large number of repetitions in order to obtain experimental results of statistical significance. However, it has been shown that the results are highly repeatable when the same droplet ensemble is used for the experiments. A droplet ensemble can be used for several days, since its coverage with silicone oil drastically reduces droplet evaporation.

Although the experimental methodology actually offers a large variety of possibilities for experimentation, in the present study only droplet ensembles deposited onto a substrate have been considered. However, the methodology also allows investigation of ice nucleation in water droplets e.g. emulsified in a carrier fluid, which may be filled into the groove in the ceramic. Since such droplets would not be in direct contact with the ceramic or any other rigid substrate, this configuration would be preferable for the investigation of the effectiveness of certain ice nucleating particles immersed into the individual water droplets of the emulsion (Whale et al. (2018); Murray et al. (2012)).

In conclusion, the present approach offers a wide variability of the experimental conditions for which both the statistics of ice nucleation and the freezing of supercooled water droplets can be examined with an excellent accuracy and repeatability of the thermal conditions. The option for imposing a well-defined electric field of arbitrary type further increases the range of possibilities. In particular, it allows investigation of physical problems never examined before.

*Video supplement.* The supplementary material contains videos of oscillating droplets for different surrounding media, namely oil (Löwe (2020b)) and air (Löwe (2020a)). The videos are used under CC BY 3.0 DE. Both videos are recorded with a frame rate of $2000$ fps and show a droplet of $V = 50\,\mu l$ at a frequency of $f = 17.6$ Hz and an electric field strength of $E = 3.68$ kV/cm.

*Author contributions.* JML and MS designed and built the experimental setup. The experiments were conducted by JML as well as students supervised by both JML and MS. VH and CT contributed valuable discussion and proof-read the manuscript. The manuscript was prepared by MS and JML with improvements of the co-authors.

*Competing interests.* The authors declare that they have no conflict of interest.

*Financial support.* The research is financially supported by Deutsche Forschungsgemeinschaft (DFG) within the Collaborative Research Centre SFB-TRR 75, Project number 84292822.



*Acknowledgements.* The authors want to thank Tim Dorau, Julia Wenzel, Patrik Weis and Moritz Hülsebrock for their contribution to the general design of the experimental setup. Furthermore, the authors thank Julian Moxter and Peter Hock for their support in developing the AC power supply and the impulse generators.





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
