# Peer review of "Experimental methodology and procedure for SAPPHIRE: a Semi-automatic APParatus for High-voltage Ice nucleation REsearch"

_Atmospheric Measurement Techniques, 2020_

## Referee Comment (RC1) · Anonymous Referee #1 · 27 Aug 2020

The manuscript by Löwe and co-authors gives an in-detailed description on the SAP-PHIRE, an instrument to study the impact of electric fields on ice nucleation in water droplets. Though it is possible that electric fields might impact ice formation in clouds and induce icing of infrastructure, it is not well studied to-date.

For experiments in SAPPHIRE, droplets of equal sizes are generated and placed on a saphhire glass sheet, and covered with silicone oil. Thereafter, the droplets are placed within a housing, where they can be exposed to subzero temperatures and different electric fields. The phase change of the droplets can thereby be monitored with two high- and low-speed cameras.

The manuscript provides a detailed discussion on the boundary conditions of the experimental setup regarding the motion and resonance frequencies of droplets in oil as compared to in air, the electric field strength distribution in the different materials (sapphire, liquid water, and oil), and a potential charge effect on the droplets. The authors demonstrate that the SAPPHIRE instrument is capable of simulating relevant conditions regarding the impact of electric fields on the freezing behaviour of droplets. Thus I recommend its publication in AMT.

The manuscript is well written and clear, and I only have the following minor comments:

Technical:

- Citation style for references in brackets should be changed to ". . . (Smith et al., 2009), see manuscript preparation guidelines for authors.

- The figures' caption text sizes seem to be too small, please double-check.

- Please insert spaces between number and unit, and units must be written exponentially , following the guidelines of AMT.

- I recommend to be consisten with either using °C or K (e.g. you note temperatures in °C, while cooling rates are given in K min-1)

General:

- Page 1, line 18: A reference for the relevance of ice nucleation in atmospheric research should be given here.

- Page, 1, lines 18 – 19: You might want to include the field of food science, where freezing processes are used to preserve food (e.g. You et al., 2020).

- Page 2, lines 33 – 34: The overview study by Kanji et al. (2017) might be suited for citation as well.

- Page 2, lines 53 – 54: Statement "Since ice nucleation in sessile water droplets may

be affected by a variety of factors such as the droplet size, their proximity to other droplets, and contamination on the substrate surface or in the droplet…" needs a reference.

- Page 3, line 59 and page 4, lines 92 – 93: It might be useful to additionally state the electrical and thermal conductivities at sub-zero temperatures, e.g. at -20°C.

- Page 4, line 104: What is the meaning of $\ddot{I}\acute{S}\_d$, as compared to $\ddot{I}\acute{S}$ (page 11, line 192)? - Page 5, line 116: Why did you introduce $\acute{a}\acute{z}\cancel{l}$?

- Page 5, lines 118 – 119: I suggest to be more quantitative here and to highlight that the temperature distribution across the sapphire substrate is 0 °C, at least using your temperature sensor.

- Page 11, line 200: What is the temperature uncertainty arising from your temperature sensor? It would be helpful to have a total temperature uncertainty at the minimum temperature, combining the sensor uncertainty and the deviation from the target temperature from the calibration.

- Page 12, Figure 8: Please introduce or the abbreviations used for the axis labels ($\ddot{I}\acute{S}\_g$, $\ddot{I}\acute{S}\_h$), and also be consistent with either using °C or K.

- Page 13, lines 234 – 235: Please specify here that, although your measurement setup allows accurate temperature control down to -34°C, that heterogeneous nucleation experiments can only be performed down to approximately -25°C, where all your droplets are frozen; otherwise this sentence might be misleading for the readers.

- Page 14, line 266: "Figure" should be abbreviated.

- Page 16, lines 316 – 318: Which value has the permittivity of oil? How much higher is it as compared to air, and how much lower as compared to water?

- Page 18, lines 350 – 351: Could you quantify to which value the electric field strength is limited to prevent the formation of a Taylor cone? And would this effect be important

for droplets in real-world, e.g. would the droplet size be reduced significantly?

- Page 19, lines 374 and following: You might also refer to the discussion of Niedermeier et al., (2011) regarding stochastic and singular ice nucleation.

- Page 20, lines 390 – 391: Please also add to the discussion that the remaining scatter of the freezing behaviours using the same sample ensemble might also be related to the method's uncertainty, e.g. temperature uncertainty, or determination of freezing events via a camera.

- Page 21, line 409: "Visualized"

- Page 21, line 412: Although your freezing apparatus might be capable of reaching temperatures down to -40°C, your droplets are freezing at temperatures warmer than $-25$°C; thus you cannot investigate any effect of an electric field on liquid droplets colder than this temperature. Thus the statement "… a temperature of $-40$ °C, may be imposed onto the droplet ensemble." Is not correct, since there would not be any liquid droplets colder than -25°C, only frozen droplets.

- Page 21 lines 429 – 430: You could also state here that this effect would rather cause an underestimation on the effect on ice nucleation.

- Page 22, line 441 – 442: This might also be related to your preparation of the samples, e.g. not preparing the droplets clean enough, such that it is not necessarily a stochastic effect.

References

Kanji, Z. A., Luis A. Ladino, Heike Wex, Yvonne Boose, Monika Burkert-Kohn, Daniel J. Cziczo, and Martina Krämer. 2017. "Overview of Ice Nucleating Particles." Meteorological Monograophs. doi: 10.1175/amsmonographs-d-16-0006.1.

Niedermeier, D., R. A. Shaw, S. Hartmann, H. Wex, T. Clauss, J. Voigtländer, and F. Stratmann. 2011. "Heterogeneous ice nucleation: exploring the transition from

stochastic to singular freezing behavior." Atmospheric Chemistry and Physics. doi: 10.5194/acp-11-8767-2011.

You, Youngsang, Taiyoung Kang, and Soojin Jun. 2020. "Control of Ice Nucleation for Subzero Food Preservation." Food Engineering Reviews. doi: 10.1007/s12393-020-09211-6.

---

## Referee Comment (RC2) · Anonymous Referee #2 · 15 Sep 2020

Review of "Experimental methodology and procedure for SAPPHIRE: a Semi-automatic APParatus for High-voltage Ice nucleation REsearch" by Löwe et al.

General comment: The SAPPHIRE is an interesting apparatus that can be used to improve the current understanding on how high-voltages and/or electric fields can impact ice nucleation. The authors were careful to evaluated the capabilities of the new apparatus; however, additional experiments are needed to proof that SAPPHIRE can actually evaluate if electric fields can affect ice nucleation. The following points need to be properly addressed before the manuscript can be accepted for its publication.

Major comments: 1. The logic of the Introduction is not the best. The authors move

back and forward with the same topic (e.g., ice nucleation). 2. The main goal of the SAPPHIRE is that it can be used to investigate the effect of high-voltages (or electric fields) on ice nucleation; however, the authors did not provide a single experiment in this direction. The provided ice nucleation results are in the absence of electric fields. How can we be sure that SAPPHIRE can actually do what this? 3. The author claim they can run heterogeneous ice nucleation experiments with their setup, but it is not mentioned what heterogeneous ice nucleation modes can be studied with the present setup and how the experiments will be performed.

Minor comments: 1. The English needs to be improved. 2. The authors are not citing correctly. This needs to be fixed along the manuscript. 3. Please change " nucleation" to "ice nucleation" along the text. 4. In the Introduction the following needs to be added: 1) What has beed reported in the literature about the potential effects of electric fields on ice nucleation? 2) Introduce the devices previously build to study this phenomena. 5. In several places the authors talk about ice nucleation without a clear distinction between heterogeneous ice nucleation and homogeneous ice nucleation. It has to be clearly stated that they are not the same. 6. P1 Line 40: Add a reference after "risks". 7. P1 Line 44: "impurities". Do the authors mean "aerosol particles"? 8. P1 Lines 43-44: How about ice supersaturation? 9. P1 Line 56: Add a reference after "sheds". 10. P1 Line 58: Add a reference after "field". 11. P2 Line 2: Add a reference after "field". 12. P2 Line 43: "contamination on". Please clarify this. 13. P3: Define PMMA 14. P3 Line 18: I suggest to use older and pionering references here. 15. P3 Line 49: What is the temperature uncertainty? 16. P3 Line 50: "heterogeneous". What heteregonueous modes can be run here? 17. P4 Line 41. "Figure 4". 18. P5 Line 32: "ice nucleation". Heterogeneous or homogeneous? 19. P6 Line 19: "Figure 7". 20. P8 Line 64: "Figure 10". 21. P9 Line 57: "Figure 11". 22. P10 Line 81: A referece is missing. 23. P11 Lines 20-26: Why are the authors talking about "heterogeneous" if these experiments were run for pure water? Did you use INPs? what type? 24. P11 Line 46: "Figure 12". 25. P11: Figure 13 is not mentioned in the main text.

---

## Author Comment (AC2) · 13 Oct 2020

We would like to thank the reviewer for the analysis of our manuscript. Please see the supplementary file for our point-by-point answer including a highlighted manuscript with all changes.

Please also note the supplement to this comment:
https://amt.copernicus.org/preprints/amt-2020-249/amt-2020-249-AC2-supplement.pdf

---

## Author Response (AR1)

**Responses to Reviewer #1**

We would like to thank the reviewer for the detailed analysis of our manuscript and we appreciate the excellent quality of his/her comments. The critical review clearly helped us to improve the manuscript. We have checked all points and revised the manuscript. In addition, we have prepared a point-by-point answer to the reviewer's comments and criticism. The revised, redlined manuscript is attached to this reply. In this reply the reviewer comments are bold-faced, while our responses are given in the plain font.

**Technical:**

**1) Citation style for references in brackets should be changed to "...(Smith et al., 2009), see manuscript preparation guidelines for authors.**

We have changed the citation style as recommended and required by the manuscript preparation guidelines. All citations are now in the following format: (Smith et al., 2009)

**2) The figures' caption text sizes seem to be too small, please double-check.**

We have labeled the figures according to the latex template provided by AMT and have not changed the font size of the labels. Consequently, we believe that the size should be correct and might only appear rather small in the preprint. We expect that this will be different in the final manuscript, which will be typeset in production. Perhaps the editor could instruct us otherwise.

**3) Please insert spaces between number and unit, and units must be written exponentially, following the guidelines of AMT.**

We have already separated our number and units by a half space. In order to make it even more clear, we have changed the half space to a full space. In addition, all units are now written exponentially as stated in the guidelines of AMT.

**4) I recommend to be consistent with either using °C or K (e.g. you note temperatures in °C, while cooling rates are given in K min-1)**

Thank you for the comment. We are using the variable $\vartheta$ to quantify temperatures in degree Celsius because this unit, even if it is not a SI-unit, is more intuitive for most people compared to Kelvin. In contrast, we are using $T$ for quantities measured in Kelvin. Our system is still consistent because the unit Kelvin is only used for temperature differences and quantities derived from temperature differences, such as the cooling rate. Such differences should always be quantified in SI-units. In addition, we want to avoid using the same variable with two different units, which is indeed not consistent. Nevertheless, we have adjusted the cooling rates according to the manuscript guidelines to the following K min$^{-1}$.

**General:**

**5) Page 1, line 18:  A reference for the relevance of ice nucleation in atmospheric research should be given here.**

We have added the following citations to prove our statement:

Cantrell, W., and A. Heymsfield, 2005: Production of Ice in Tropospheric Clouds: A Review. Bull. Amer. Meteor. Soc., 86, 795–808, https://doi.org/10.1175/BAMS-86-6-795

Kanji, Z. A., L. A. Ladino, H. Wex, Y. Boose, M. Burkert-Kohn, D. J. Cziczo, and M. Krämer, Overview of Ice Nucleating Particles. *Meteor. Monogr.*, 2017; 58 1.1–1.33. doi: https://doi.org/10.1175/AMSMONOGRAPHS-D-16-0006.1

**6) Page 1, lines 18 – 19:  You might want to include the field of food science, where freezing processes are used to preserve food (e.g. You et al., 2020).**

Thank you for highlighting this interesting work. The reviewer is correct that ice nucleation is also investigated in food science. Hence, we have adapted the introduction and added the reference:

"Ice nucleation and the involved physics are present in many technical fields such as atmospheric research, the icing of aircraft, ships or structures, as well as in the food sciences (Cantrell and Heymsfield, 2005; Kanji et al., 2017; Cebeci and Kafyeke, 2003;Makkonen, 1987; Szilder et al., 2002; You et al., 2020)."

**7) Page 2, lines 33 – 34:  The overview study by Kanji et al.  (2017) might be suited for citation as well.**

Thank you for mentioning this excellent overview from Kanji et al.. We have added the citation as recommended.

**8) Page 2, lines 53 – 54: Statement "Since ice nucleation in sessile water droplets may affected  by  a variety  of  factors  such  as  the  droplet size,  their  proximity  to  other  droplets, and contamination on the substrate surface or in the droplet..." needs a reference.**

We have added several citations to underline our statement.

Influence of droplet size:
1) Bigg, E. K. (1953). The supercooling of water. Proceedings of the Physical Society. Section B, 66(8), 688.
2) Heverly, J. R. (1949), Supercooling and crystallization, *Eos Trans. AGU*, 30( 2), 205– 210,

Influence of proximity of droplets:
1) Storelvmo, T., & Tan, I. (2015). The Wegener–Bergeron–Findeisen process—Its discovery and vital importance for weather and climate. *Meteor. Z*, *24*, 455-461.

Influence of substrate surface:
1) Campbell, J. M., Meldrum, F. C., & Christenson, H. K. (2015). Is ice nucleation from supercooled water insensitive to surface roughness? *The Journal of Physical Chemistry C*, *119*(2), 1164-1169.
2) Holden, M. A., Whale, T. F., Tarn, M. D., O'Sullivan, D., Walshaw, R. D., Murray, B. J., ... & Christenson, H. K. (2019). High-speed imaging of ice nucleation in water proves the existence of active sites. Science advances, 5(2), eaav4316.

Influence of contaminations in the droplet:
1) Hoose, C., & Möhler, O. (2012). Heterogeneous ice nucleation on atmospheric aerosols: a review of results from laboratory experiments. *Atmos. Chem. Phys*, *12*(20), 9817-9854.

**9) Page 3, line 59 and page 4, lines 92 – 93: It might be useful to additionally state the electrical and thermal conductivities at sub-zero temperatures, e.g. at -20◦C.**

We have added the thermal conductivity of Shapal HI-M Soft for a temperature of -25 °C.
"The ceramic body serving as the specimen holder is made of Shapal Hi-M Soft, which exhibits a high thermal conductivity (92 W (m K)$^{-1}$ at 25 °C and $\approx 95$ W (m K)$^{-1}$ at -25 °C) and ensures a precise and fast control of the temperature of the droplets."

Unfortunately, we do not have any information regarding the electrical conductivity of the high purity water at sub-zero temperatures.

**10) Page 4, line 104: What is the meaning of ϊ̈S_d, as compared to ϊ̈S (page 11, line192)? - Page 5, line 116: Why did you introduce á´zł?**

There is no difference between $\vartheta_d$ and $\vartheta$, both variables describe the droplet temperature. To be consistent we have removed the index of $\vartheta_d$ on page 4, line 104.
We have introduced $\dot{T}$ to assign a variable for the cooling rate, but the question of the reviewer is legitimate, because we have used the term before without any definition. Now, we have explained the variable upon its first occurrence on page 4. Furthermore, we consistently used the variable throughout the entire manuscript to specify a specific cooling rate.

**11) Page 5, lines 118 – 119: I suggest to be more quantitative here and to highlight that the temperature distribution across the sapphire substrate is 0◦C, at least using your temperature sensor.**

We have added the following sentences to be more quantitative:
"Especially, the temperature distribution on the sapphire is very homogeneous and yields a vanishing deviation of temperature with respect to the reference ($\Delta T \approx 0$ K), at least according to the temperature sensor being used. Furthermore, the temperature of the ceramic body is about $\Delta T \approx -3.8$ K lower compared to the sapphire. Only the temperature of the PMMA cover deviates significantly ($\Delta T \approx 19$ K) from the reference."

**12) Page 11, line 200: What is the temperature uncertainty arising from your temperature sensor? It would be helpful to have a total temperature uncertainty at the minimum temperature, combining the sensor uncertainty and the deviation from the target temperature from the calibration.**

We have added the accuracy of the temperature sensor:
"A fiber optical measurement system (Fiso FOT-L-SD) with a measurement accuracy of $\delta T = \pm 1$ K is used for the continuous temperature measurement inside the ceramic block, since it is a non-invasive measurement method with respect to the electric field."

Furthermore, we have added the following sentence to specify the total temperature uncertainty as suggested:
"The total temperature uncertainty arising from the calibration and the accuracy of the sensor amounts to $\pm 1.46$ K for a target temperature of -34 °C."

**13) Page 12, Figure 8: Please introduce or the abbreviations used for the axis labels(ϊ̈S_g, ϊ̈S_h), and also be consistent with either using °C or K.**

We have introduced the variables used in the caption of Fig. 8 into the text:
"For that, the temperature inside the ceramic block, $\vartheta_h$, is controlled and the temperature on the sapphire substrate, $\vartheta_g$, is continuously measured to determine its dependence on the cooling rate and the current temperature of the ceramic block."

In addition, we adjusted the caption of Fig. 8:

"Example data obtained during temperature calibration. a) and b) Sapphire substrate temperature $\vartheta_g$ as a function of the ceramic temperature $\vartheta_h$ for $\dot{T} = 3$ K min$^{-1}$ and $\dot{T} = 5$ K min$^{-1}$. c) Difference between the sapphire substrate temperature $\vartheta_g$ and the ceramic temperature $\vartheta_h$ depending on the latter for $\dot{T} = 3$ K min$^{-1}$ and $\dot{T} = 5$ K min$^{-1}$. e) Test of the temperature calibration $\dot{T} = 5$ K min$^{-1}$ : difference between the indirectly measured sapphire substrate temperature and the target temperature $\vartheta_{target}$ prescribed by the Peltier controller in order to follow the temperature ramp, depending on the target temperature itself."

**14) Page 13, lines 234 – 235: Please specify here that, although your measurement setup allows accurate temperature control down to -34 °C, that heterogeneous nucleation experiments can only be performed down to approximately -25∘C, where all your droplets are frozen; otherwise this sentence might be misleading for the readers.**

We have added a sentence to clarify our statement:
"In general, heterogeneous nucleation is only investigated down to $\vartheta \approx -25$ °C, because all droplets are already frozen at this temperature for the given boundary conditions and test fluid array."

**15) Page 14, line 266: "Figure" should be abbreviated.**

We have abbreviated the word Figure, which is now shown as "Fig.".

**16) Page 16, lines 316 – 318: Which value has the permittivity of oil?  How much higher is it as compared to air, and how much lower as compared to water?**

We have added the relative permittivities of air and oil. The permittivity of water is already given in the text.
"The numerically obtained (Comsol Multiphysics) field distribution around two droplets deposited onto a sapphire substrate and surrounded by air ($\varepsilon = 1$) and oil ($\varepsilon \approx 2.8$) respectively, is shown for example in Fig. 11."

and

"The small difference in permittivity between air ($\varepsilon = 1$) and oil ($\varepsilon = 2.8$) is insignificant compared to the permittivity of water ($\varepsilon = 80$), resulting in highly comparable electric field distributions."

**17) Page 18, lines 350 – 351: Could you quantify to which value the electric field strength is limited to prevent the formation of a Taylor cone? And would this effect be important for droplets in real-world, e.g. would the droplet size be reduced significantly?**

The electric field strength to form a Taylor cone depends on the type of the electric field (alternating, constant or transient electric field) and we expect that it might also depend on the substrate and on the droplet size. Hence, determining a threshold of the electric field is difficult.
Nevertheless, the effect is important for real-world applications, like high voltage insulators. The generation of tiny droplets is a continuous and periodic process as long as the electric field is applied. Hence, a sessile droplet on a high-voltage insulator exposed to an electric field might continuously generate tiny droplets, resulting in a significant decrease of the volume over time. In this case, the deformation and spreading of the droplet on the surface might promote the separation of tiny droplets. The size distribution of the sessile droplets and the distance between the individual droplets is important for the generation of unwanted partial discharges, which deteriorate the surface. As the number of droplets on the surface increase,  the distance between them decreases and the probability

of partial discharges increase. However, we have not revised the text because we think that the occurrence of a Taylor cone is only a minor side effect in our experiments and droplets which are affected by the Taylor cones are excluded from the analysis. Moreover, a detailed analysis of the phenomenon requires an extensive study of the different influencing factors, which is more suitable for manuscript focusing on results rather than the methodology.

**18) Page 19, lines 374 and following:  You might also refer to the discussion of Niedermeier et al., (2011) regarding stochastic and singular ice nucleation.**

Thank you very much for this valuable reference. We have added a citation to this work with the following text:
"Nevertheless, nucleation is a stochastic process resulting in a natural variance of the results, even for perfectly constant conditions (Vali and Stansbury, 1966; Niedermeier et al., 2011). As reported by Niedermeier et al. (2011) the observed behavior might significantly depend on the chosen time scale and thus, the stochastic behavior might be masked."

**19) Page 20, lines 390 – 391: Please also add to the discussion that the remaining scatter of the freezing behaviours using the same sample ensemble might also be related to the method's uncertainty, e.g.  temperature uncertainty, or determination of freezing events via a camera.**

The reviewer is correct in his/her assessment of the situation and we have revised the sentence to read:
"The remaining small scatter of the survival curves might be attributed to the stochastic nature of nucleation or the method uncertainty like the uncertainty of the temperature measurement or the accuracy of the optical detection of the freezing events."

**20) Page 21, line 409: "Visualized"**

Thank you, we have corrected the typo.

**21) Page 21, line 412:  Although your freezing apparatus might be capable of reaching temperatures down to -40◦C, your droplets are freezing at temperatures warmer than –25◦C; thus you cannot investigate any effect of an electric field on liquid droplets colder than this temperature.  Thus the statement "...a temperature of –40◦C, maybe imposed onto the droplet ensemble."  Is not correct, since there would not be any liquid droplets colder than -25◦C, only frozen droplets.**

Thank you for the comment, which indeed highlights a potential misunderstanding. We have added the following sentence to be unambiguous:

"Any kind of temperature profile – constant temperature, constant cooling rate, or any other transient temperature evolution – with rates of change of the temperature of up to 5 K min$^{-1}$ down to a temperature of -40 °C may be imposed on the droplet ensemble. Nevertheless, the nucleation temperature of the individual droplets is typically higher, depending on the boundary conditions, like the substrate or the cleanness of the droplet ensemble."

As you can see, our experimental setup is capable to cool down a droplet ensemble to a temperature of -40 °C at a cooling rate of 5 K/min. Even if individual droplets freeze at a higher temperature, the complete ensemble can be cooled with this defined rate. The term "droplet ensemble" does not specify whether the droplets are liquid or already frozen. The presented data only shows liquid droplets down to temperature of -25 °C, but this statement is only valid for the given boundary conditions (substrate, droplet ensemble…). So, our text now clarifies that the nucleation of the individual droplets may occur at higher temperatures, depending on the boundary conditions.

**22) Page 21 lines 429 – 430: You could also state here that this effect would rather cause an underestimation on the effect on ice nucleation.**

We have added a sentence to the conclusion to highlight this point:
"Therefore, using silicone oil as surrounding fluid would lead to an underestimation of the influence of the electric field on ice nucleation compared to a droplet surrounded by air."

**23) Page 22, line 441 – 442: This might also be related to your preparation of the samples, e.g. not preparing the droplets clean enough, such that it is not necessarily a stochastic effect.**

We are not quite sure about the meaning of the reviewer's comment, but we believe that the word "stochastic" is confusing. The reviewer is correct, that it is not necessarily a stochastic effect with respect to nucleation. Hence, we have removed the word and changed the sentence from

"Although the entire method for sample preparation aims at maximizing the repeatability of the experiments, the test experiments indicate a significant stochastic effect of sample preparation on ice nucleation."

to

"Although the entire method for sample preparation aims at maximizing the repeatability of the experiments, the test experiments indicate a significant effect of sample preparation on ice nucleation."

**Responses to Reviewer #2**

We would like to thank the reviewer for the analysis of our manuscript. Unfortunately, according to the page and line references used in the review it appears that the reviewer refers to the highlighted version which we handed in during resubmission of our manuscript; however, this version is not available for all. Hence, the actual line numbers differ from the preprint. Nevertheless, we carried out the revision, addressing all the points of the reviewer. In addition, we have prepared a point-by-point answer to the reviewer's comments and criticism.

The revised manuscript is attached to this reply. Note that the changes made according to the suggestions of reviewer #2 are colored in green, while changes made according to reviewer #1 are colored in red. In this reply the reviewer comments are bold-faced, while our responses are given in the plain font. Further editorial and language changes were made to the manuscript; however, these are too numerous to mark separately.

**Major comments:**

1. **The logic of the Introduction is not the best. The authors move back and forward with the same topic (e.g., ice nucleation).**
   Although the reviewer is not too specific with this remark, we of course appreciate that an introduction should contextualize the topic even for lay readers and at the same time point out missing gaps in our knowledge which the study attempts to fill. We felt that we had done this rather succinctly; however, in response to this remark, we have added some sentences to the introduction to help better weave the different parts together.
   - Our introduction begins with the general importance of ice nucleation for several applications with some examples. Subsequently, two specific applications of ice nucleation, namely ice particle formation in clouds and icing of high-voltage insulators are described. These applications are given as examples where an electric field can have an influence on ice nucleation. Although these two examples are quite diverse in potential influencing factors, they can both be well investigated using our experimental setup. Furthermore, we mention literature dealing with the influence of electric fields on ice nucleation, and because the results of these previous studies are in many respects contradictory, we can explicitly identify the knowledge gap and research needs. This provides motivation for our work. The introduction concludes with a short overview of the manuscript.
   Of course if the reviewer wants to be more specific, we are happy to make further modifications.

2. **The main goal of the SAPPHIRE is that it can be used to investigate the effect of high-voltages (or electric fields) on ice nucleation; however, the authors did not provide a single experiment in this direction. The provided ice nucleation results are in the absence of electric fields. How can we be sure that SAPPHIRE can actually do what this?**
   The reviewer is correct that no results are presented in this manuscript which demonstrate the influence of an electric field on ice nucleation. This is intentional and the reason is because the aim is to focus solely on the experimental methodology and not on systematic or comprehensive results. We have formerly published another article in which the influence of the electric field on ice nucleation is demonstrated and this article has been cited several times in the manuscript. Thus we are very confident that the apparatus fulfils its goals. (Please refer to: Löwe, J.-M., Schremb, M., Hinrichsen, V., and Tropea, C.: Ice Nucleation in the Presence of Electric Fields: An Experimental Study, SAE Technical Paper Series, SAE International 400 Commonwealth Drive, Warrendale, PA, United States, https://doi.org/10.4271/2019-01-2020, 2019.)
   Adding more results to the present manuscript would unnecessarily increase its length (which is already quite long), especially since a convincing proof of the electric field influence requires a large number of experiments and a very detailed analysis. This detailed analysis considers several influencing factors like the electric field strength, type of electric field or frequency of the electric

field and multiple repetitions of each condition to obtain statistical significance. In the opinion of the authors, such a comprehensive description of these experiments would be detrimental to the focus of the present manuscript.

**3. The author claim they can run heterogeneous ice nucleation experiments with their setup, but it is not mentioned what heterogeneous ice nucleation modes can be studied with the present setup and how the experiments will be performed.**

The reviewer is correct that we have not mentioned any heterogeneous ice nucleation modes; the manuscript only addresses the experimental setup and the boundary conditions. We exemplary show how we prepare sessile droplets to investigate contact freezing, but also different methods of sample preparation are possible. The investigated ice nucleation mode mainly depends on the sample preparation because, for example, adding particles to the fluid clearly changes the sample preparation. Because the present work aims at presenting the capabilities, and more important the boundary conditions of our experimental setup, we believe that it is not necessary to specify all of the ice nucleation modes which can be studied using SAPPHIRE. The functionality of the experimental setup and the boundary conditions, like the temperature, electric field and observation system, are fully independent of which ice nucleation modes are being studied.

Nevertheless, because we use high purity water implies that the water does not contain any additional particles. Furthermore, the deposition on the sapphire glass sheet always leads to contact freezing. Hence, we believe that there is no danger of confusion. For a manuscript focusing on ice nucleation results, of course a more detailed analysis of the ice nucleation modes would be necessary, as the reviewer suggests.

**Minor comments:**

**1. The English needs to be improved.**
Thanks for the comment, which motivated us to again proof read the manuscript. Note, that one of the authors is a native speaker.

**2. The authors are not citing correctly. This needs to be fixed along the manuscript.**
We have changed the citation style according to the AMT guidelines, which now appears in the format: Smith (2009).

**3. Please change "nucleation" to "ice nucleation" along the text.**
We agree with this suggestions and have implemented it throughout the manuscript, wherever suitable.

**4. In the Introduction the following needs to be added:**
  **1) What has been reported in the literature about the potential effects of electric fields on ice nucleation?**
  **2) Introduce the devices previously build to study this phenomena.**
Thanks also for this remark, which has led us to include some additional text and citations. At the same time, we have attempted to underline the fact that some previous studies have resulted in certain contradictory results. In any case, we have attempted to find previous work on many influencing factors, including the type of the electric field or field strength.
The added references are listed below.

1) Acharya, Palash V., and Vaibhav Bahadur. "Fundamental interfacial mechanisms underlying electrofreezing." *Advances in colloid and interface science* 251 (2018): 26-43.
2) Ma, Yahong, et al. "Manipulating ice crystallization of 0.9 wt.% NaCl aqueous solution by alternating current electric field." *Applied Physics Letters* 102.18 (2013): 183701.
3) Salt, R. W. "Effect of electrostatic field on freezing of supercooled water and insects." *Science* 133.3451 (1961): 458-459.

4) Wei, Sun, et al. "Effects of dipole polarization of water molecules on ice formation under an electrostatic field." *Cryobiology* 56.1 (2008): 93-99.
5) Petersen, Ansgar, et al. "A new approach for freezing of aqueous solutions under active control of the nucleation temperature." *Cryobiology* 53.2 (2006): 248-257.

Nonetheless, we have refrained from being too comprehensive in reviewing all previous apparatus, since most are very different from our own concept and none offer comparable capabilities and repeatability.

**5. In several places the authors talk about ice nucleation without a clear distinction between heterogeneous ice nucleation and homogeneous ice nucleation. It has to be clearly stated that they are not the same.**
This is a very good point raised by the reviewer, since the two types of nucleation stem from different mechanisms of nucleation. Fundamentally, both heterogeneous and homogeneous nucleation can be investigated using our experimental setup. The actual type of nucleation that occurs will clearly depend on sample preparation (sessile droplets vs. emulsified droplets). Because we described the sample preparation of sessile water droplets in detail, we felt that it would be obvious that we are dealing with heterogeneous nucleation. However, because emulsified droplets could be used in our experimental setup, also homogeneous nucleation could be studied.

Therefore, in response to the reviewer's comment, when we mention this application with emulsified droplets we used the term "homogeneous nucleation" to be very precise. In several cases we use the more general term "ice nucleation" when both heterogeneous and homogeneous nucleation are meant. We have avoided stating that heterogeneous and homogeneous nucleation are the same and explicitly mentioned homogeneous nucleation if it is meant.

**6. P1 Line 40: Add a reference after "risks".**
Good point. We added the following citations:

1) Baars, Woutijn J., Ronald O. Stearman, and Charles E. Tinney. "A Review on the Impact of Icing on Aircraft Stability and Control." *Journal of Aeroelasticity and Structural Dynamics* 2.1 (2010).
2) Farzaneh, Masoud. "Ice accretions on high–voltage conductors and insulators and related phenomena." *Philosophical Transactions of the Royal Society of London. Series A: Mathematical, Physical and Engineering Sciences* 358.1776 (2000): 2971-3005.

**7. P1 Line 44: "impurities". Do the authors mean "aerosol particles"?**
The authors appreciate this suggestion to be more precise; thus we have changed the present text from

"During ice particle formation in clouds, multiple influences like supercooling, impurities and external electric field are present and might influence ice nucleation (Vali, 1996;Cantrell and Heymsfield, 2005; Pruppacher and Klett, 2010)."

to

"During ice particle formation in clouds, multiple influences like supercooling, solid aerosol particles (like e.g. dust particles or Volcanic ash) and an external electric field are present and might influence ice nucleation (Vali, 1996; Cantrell and Heymsfield, 2005; Pruppacher and Klett, 2010; Murray et al., 2012)."

In addition we added the following reference:

1) Murray, B. J., et al. "Ice nucleation by particles immersed in supercooled cloud droplets." *Chemical Society Reviews* 41.19 (2012): 6519-6554.

**8. P1 Lines 43-44: How about ice supersaturation?**
We agree with the reviewer on this point that also ice supersaturation can influence the formation of ice particles in clouds. Our present list of influencing factors does not claim to be all inclusive, but names only the influencing factors important for this study. Nevertheless, we added ice supersaturation to the list.

During ice particle formation in clouds, multiple influences like supercooling, solid aerosol particles (like e.g. dust particles or Volcanic ash), ice supersaturation and an external electric field are present and might influence ice nucleation (Vali, 1996; Cantrell and Heymsfield, 2005; Pruppacher and Klett, 2010; Murray et al., 2012)."

**9. P1 Line 56: Add a reference after "sheds".**
We have added the following citation:
1) Farzaneh, M.: Atmospheric Icing of Power Networks, Springer, 1. edn., 2008.

**10. P1 Line 58: Add a reference after "field".**
The sentence in question does not strictly claim that homogeneous and heterogeneous nucleation is affected by an electric field, but leaves this as an open question which needs to be answered:

"Both homogeneous as well as heterogeneous ice nucleation might be influenced by the electric field."

Due to the fact that published results revealed significantly different and in some cases contradictory outcomes, the sentence cannot be confirmed with a reference and the contradictory results are already referenced in the manuscript. We state that the electric field might have an influence. In fact, this is the reason why we developed this experimental setup to provide reliable results to prove such statements.

**11. P2 Line 2: Add a reference after "field".**
The sentence

"Whether or not ice particles are formed in clouds or on the surface of e.g. an insulator, is controlled by ice nucleation in the water droplets, which may be affected by the electric field."

again poses an open question which can and should be investigated with the developed experimental setup. Furthermore, it is intended to point out that the influence of the electric field on ice nucleation is not yet completely clear. As already mentioned in point 10, we believe that adding a reference has no benefit because the most relevant ones in this context are already referenced in the manuscript.

**12. P2 Line 43: "contamination on". Please clarify this.**
Good point.
Any kind of impurities on the surface might promote heterogeneous nucleation. In our case the contamination might be small dust particles, because even a careful and clean sample preparation cannot rule out local contamination of e.g. non-visible ice nucleate particles.

Therefore, to be more precise we have changed the sentence from:
"Since ice nucleation in sessile water droplets may affected by a variety of factors such as the droplet size, their proximity to other droplets, and contamination on the substrate surface or in the droplet, special attention is required during sample preparation in order to control boundary conditions as accurately and repeatability as possible.."

to

"Since ice nucleation in sessile water droplets may be affected by a variety of factors such as the droplet size (Bigg, 1953; Heverly, 1949), their proximity to other droplets (Storelvmo and Tan, 2015), and contamination of impurities on the substrate surface (Campbell et al., 2015; Holden et al., 2019) or in the droplet (Hoose and Möhler, 2012) promoting ice nucleation, special attention is required during sample preparation in order to control the boundary conditions as accurately and repeatable as possible."

**13. P3: Define PMMA**
To define PMMA we have modified the sentence in question from

"The experimental setup is enclosed in Styrofoam to minimize heat transfer from the surroundings."

to

"The experimental setup is enclosed in Styrofoam with embedded PMMA (polymethylmethacrylate) sheets to minimize heat transfer from the surroundings and to ensure optical access to the glass sheet."

**14. P3 Line 18: I suggest to use older and pioneering references here.**
Thanks, we have added an additional reference:

1) Findeisen, W., Volken, E., Giesche, A. M., & Brönnimann, S. (2015). Colloidal meteorological processes in the formation of precipitation. Meteorologische Zeitschrift, 24(4), 443-454.

The added reference is the translated version of the original paper "Die kolloidmeteorologischen Vorgänge bei der Niederschlagsbildung" (Colloidal meteorological processes in the formation of precipitation) by Walter Findeisen that was published in 1938. We believe it is more useful to use the translated version rather than the German version, because the number of people understanding German is limited.

**15. P3 Line 49: What is the temperature uncertainty?**
The temperature uncertainty is analyzed in detail in section 2.3.2 "Temperature measurement". The calibration yields a maximum deviation of 0.46 K from the desired temperature and our measurement sensor has an accuracy of $\pm 1$K.
To be more precise we have added the total temperature uncertainty by inserting the following sentence in section 2.3.2 "Temperature measurement":

"The total temperature uncertainty arising from the calibration and the accuracy of the sensor amounts to 1.46 K for a target temperature of -34 °C."

**16. P3 Line 50: "heterogeneous". What heterogeneous modes can be run here?**
The present sentence

"Therefore, the entire temperature range relevant for both heterogeneous and homogeneous ice nucleation is covered with this setup (Langham et al., 1958; Franks, 1982)."

states that temperatures low enough to investigate homogeneous nucleation can be reached with the experimental setup. Due to the fact that the nucleation temperature of heterogeneous nucleation is higher compared to homogeneous nucleation, both nucleation mechanisms can be investigated. Hence, the investigation is independent of the different heterogeneous modes. Therefore, it does not appear necessary to describe the modes here.

We might be able to study different heterogeneous modes depending on the sample preparation. In general, we only observe contact freezing, but by adding some additional INP we might also be able to investigate immersed freezing. As described in the manuscript, we are only using high purity water and so no additional INP are mentioned. In fact, the water itself might contain particles which lead to immersed nucleation, but we assume that the concentration is negligible. Previous experiments (Schremb, Markus, and Cameron Tropea. "Solidification of supercooled water in the vicinity of a solid wall." *Physical Review E* 94.5 (2016): 052804.) revealed that ice nucleation of high purity water is almost always initiated at the substrate, which substantiates our assumption.

**17. P4 Line 41. "Figure 4".**
We have changed "As shown in the figure" to "As shown in Fig. 4".

**18. P5 Line 32: "ice nucleation". Heterogeneous or homogeneous?**
Our observation system comprises a high or low speed camera and a magnification lens with coaxial illumination. Ice nucleation is detected optically due to the changed light refraction of ice compared to water, as shown in Fig. 1. This optical change can be observed during ice nucleation of sessile and emulsified droplets. Hence, the observation system of our experimental setup is capable to capture ice nucleation in general, which is indicated by the general term "ice nucleation" without any specification. Since nucleation in the present case takes place well above the homogeneous nucleation temperature of water, we obviously see the results of heterogeneous nucleation. We feel that this point should be obvious to readers in this field.

**19. P6 Line 19: "Figure 7".**
We have changed "As shown in the figure" to "As shown in Fig. 7".

**20. P8 Line 64: "Figure 10".**
We have changed "As shown in the figure" to "As shown in Fig. 10".

**21. P9 Line 57: "Figure 11".**
We have changed "As shown in the figure" to "As shown in Fig. 11".

**22. P10 Line 81: A reference is missing.**
Thanks for the comment. Unfortunately, the reference is missing in the reviewer's version. The published preprint version already contains the correct citation (Löwe, J.-M., Schremb, M., Hinrichsen, V., and Tropea, C.: Ice Nucleation in the Presence of Electric Fields: An Experimental Study, SAE Technical Paper Series, SAE International 400 Commonwealth Drive, Warrendale, PA, United States, https://doi.org/10.4271/2019-01-2020, 2019).

**23. P11 Lines 20-26: Why are the authors talking about "heterogeneous" if these experiments were run for pure water? Did you use INPs? what type?**
The exemplary results are performed using sessile water droplets of high purity water. The water droplets are placed on a sapphire glass sheet by the drop-on-demand generator as described in the manuscript. Hence, the individual droplets are in contact with the sapphire glass, which promotes the heterogeneous nucleation. Ice nucleation is observed at specific spots on the sapphire substrate, which cannot be caused by homogeneous nucleation. This can also be seen by the ice nucleation temperature, which is significantly higher than the typical nucleation temperature for homogeneous nucleation. We did not add any INP to the water. Nevertheless, the high purity water might still contain some INP of unknown type, which might also be a result of the sample preparation being not perfectly clean. The influence of these particles is assumed to be negligible, as already stated under point 16.

**24. P11 Line 46: "Figure 12".**
We have changed "As shown in the figure" to "As shown in Fig. 12".

**25. P11: Figure 13 is not mentioned in the main text.**
Thank you for the mentioning the missing crosslink. We have added the following sentence to mention the figure inside the text.

[revised manuscript text omitted]

---

## Referee Report (RR1)

I thank the authors for taking into account most of my comments. However, my second MAJOR (and the most important) comment was not addressed.

Reviewer comment: The main goal of the SAPPHIRE is that it can be used to investigate the effect of high-voltages (or electric fields) on ice nucleation; however, the authors did not provide a single experiment in this direction. The provided ice nucleation results are in the absence of electric fields. How can we be sure that SAPPHIRE can actually do what this?

Author's response: The reviewer is correct that no results are presented in this manuscript which demonstrate the influence of an electric field on ice nucleation. This is intentional and the reason is because the aim is to focus solely on the experimental methodology and not on systematic or comprehensive results. We have formerly published another article in which the influence of the electric field on ice nucleation is demonstrated and this article has been cited several times in the manuscript. Thus we are very confident that the apparatus fulfils its goals. (Please refer to: Löwe, J.-M., Schremb, M., Hinrichsen, V., and Tropea, C.: Ice Nucleation in the Presence of Electric Fields: An Experimental Study, SAE Technical Paper Series, SAE International 400 Commonwealth Drive, Warrendale, PA, United States, https://doi.org/10.4271/2019-01-2020, 2019.) Adding more results to the present manuscript would unnecessarily increase its length (which is already quite long), especially since a convincing proof of the electric field influence requires a large number of experiments and a very detailed analysis. This detailed analysis considers several influencing factors like the electric field strength, type of electric field or frequency of the electric field and multiple repetitions of each condition to obtain statistical significance. In the opinion of the authors, such a comprehensive description of these experiments would be detrimental to the focus of the present manuscript.

Although it is true that in Löwe et al. (2019) ice nucleation experiments at a cooling rate of 5 K/min for a constant electric field of 0 kV/cm, 2.93 kV/cm, and 4.68 kV/cm are provided, the Löwe et al. (2019) results indicate that constant electric fields has a negligible effect on heterogeneous ice nucleation (Figure 9). Given that the present study introduced the possibility that SAPPHIRE has to use alternating and transient electric fields (in addition to constant electric fields), I am convinced that the authors need to show how alternating and transient electric fields can impact on heterogeneous ice nucleation.

I do not think that claiming that the paper is already long (14 pages) is a good answer for not including these important and sort of mandatory results.

Note that when the original manuscript was rejected, the following comment was provided to the authors:

Reviewer comment: One of the main novel aspects of the submitted manuscript is the possibility to use alternating or transient electric fields, however no experimental results are presented to demonstrate the effect. This should be added in a revised manuscript.

Author's response: On the first point we would prefer to offer the simple rebuttal that further results would significantly overload the manuscript, if they were presented in an adequate manner. Therefore, we prefer not to add these result.

Finally, when the authors resubmitted the manuscript they claimed the following:

In our SAE paper we focused on the influence of a constant electric field on ice nucleation and discussed the main effects and the physical mechanism in detail. Why the author do not want to do the same here for transient and alternating electric fields?

In conclusion, the same IMPORTANT request was made twice but the authors are not willing to add these results claiming that the manuscript is "too" long. One possibility to reduce the length of the manuscript is to remove Figures 2 and 12 as they are copied from Löwe et al. (2019). Those Figures can go to the supplementary material, if needed.

---

## Author Response (AR2)

**Responses to Reviewer #2**

We would like to thank the reviewer for the further review of our manuscript. We have prepared a point-by-point answer to the reviewer's comments and criticisms. The revised manuscript is attached to this reply and all changes made are colored in red. In this reply the reviewer comments are bold-faced, while our responses are given in plain font.

**Although it is true that in Löwe et al. (2019) ice nucleation experiments at a cooling rate of 5 K/min for a constant electric field of 0 kV/cm, 2.93 kV/cm, and 4.68 kV/cm are provided, the Löwe et al. (2019) results indicate that constant electric fields has a negligible effect on heterogeneous ice nucleation (Figure 9). Given that the present study introduced the possibility that SAPPHIRE has to use alternating and transient electric fields (in addition to constant electric fields), I am convinced that the authors need to show how alternating and transient electric fields can impact on heterogeneous ice nucleation.**
**I do not think that claiming that the paper is already long (14 pages) is a good answer for not including these important and sort of mandatory results.**

**Note that when the original manuscript was rejected, the following comment was provided to the authors:**

> **Reviewer comment: One of the main novel aspects of the submitted manuscript is the possibility to use alternating or transient electric fields, however no experimental results are presented to demonstrate the effect. This should be added in a revised manuscript.**

> **Author's response: On the first point we would prefer to offer the simple rebuttal that further results would significantly overload the manuscript, if they were presented in an adequate manner. Therefore, we prefer not to add these result.**

**Finally, when the authors resubmitted the manuscript they claimed the following:**

> **In our SAE paper we focused on the influence of a constant electric field on ice nucleation and discussed the main effects and the physical mechanism in detail.** **Why the author do not want to do the same here for transient and alternating electric fields?**

**In conclusion, the same IMPORTANT request was made twice but the authors are not willing to add these results claiming that the manuscript is "too" long. One possibility to reduce the length of the manuscript is to remove Figures 2 and 12 as they are copied from Löwe et al. (2019). Those Figures can go to the supplementary material, if needed.**

The authors were somewhat reserved about adding the results desired by the reviewer for several reasons, but we can also see that some sample results may be beneficial, especially given the reaction of the reviewer to our omission of such measurements. Our main concern was less the length of the manuscript, but the fact that any study of nucleation must be very carefully performed; including variation of numerous parameters to be sure that cause-and-effect is properly interpreted. Showing only singular sample measurements could easily be misleading. We would prefer to present such comprehensive studies in a separate article, avoiding any ambiguity of interpretation.

Nevertheless, understanding that sample measurements may make the usefulness of the SAPPHIRE apparatus more convincing we have added some results, indicating that an alternating electric field can systematically and significantly influence the nucleation. These additions are copied below. However, we have refrained from entering an in-depth discussion of the physical mechanisms involved, since this requires substantially more results to offer conclusive arguments.

In this manner, we hope that we have met the reviewer's concerns adequately.

We added the following paragraph:

[revised manuscript text omitted]